# Chromatin regulates IL-33 release and extracellular cytokine activity

Jared Travers [1], Mark Rochman [1], Cora E. Miracle[1], Jeff E. Habel [1], Michael Brusilovsky[1], Julie M. Caldwell [1], Jeffrey K. Rymer[1] & Marc E. Rothenberg [1]

IL-33 is an epithelium-derived, pro-inflammatory alarmin with enigmatic nuclear localization and chromatin binding. Here we report the functional properties of nuclear IL-33. Over-expression of IL-33 does not alter global gene expression in transduced epithelial cells. Fluorescence recovery after photobleaching data show that the intranuclear mobility of IL-33 is ~10-fold slower than IL-1α, whereas truncated IL-33 lacking chromatin-binding activity is more mobile. WT IL-33 is more resistant to necrosis-induced release than truncated IL-33 and has a relatively slow, linear release over time after membrane dissolution as compared to truncated IL-33 or IL-1α. Lastly, IL-33 and histones are released as a high-molecular weight complex and synergistically activate receptor-mediated signaling. We thus propose that chromatin binding is a post-translational mechanism that regulates the releasability and ST2-mediated bioactivity of IL-33 and provide a paradigm to further understand the enigmatic functions of nuclear cytokines.

[1] Division of Allergy and Immunology, Cincinnati Children's Hospital Medical Center, 3333 Burnet Avenue, Cincinnati, OH 45229-3039, USA. These authors contributed equally: Jared Travers, Mark Rochman, Cora E. Miracle. Correspondence and requests for materials should be addressed to M.E.R. (email: Marc.Rothenberg@cchmc.org)

Cytokines mediate cellular communication through activation of surface receptors upon extracellular release. A classic cytokine contains a leader peptide sequence that mediates either immediate extracellular secretion or storage in cytoplasmic secretory granules for release after cellular activation[1]. However, a subset of cytokines, including interleukin 1 (IL-1) family members and high mobility group box 1 (HMGB1), lack leader peptide sequences and instead are localized to the nucleus[2]. Amongst nuclear cytokines, much attention has been focused on IL-33, an IL-1 family member expressed by mucosal epithelial cells[3,4], because it is a potent alarmin, capable of initiating acute inflammation and priming for type 2 immune responses[5,6]. Alarmins are passively released from necrotic cells upon infection or tissue injury or are rapidly secreted by stimulated leukocytes and epithelia. Amongst alarmins, IL-33 is relatively unique in that it primes for allergic responses through its receptor, suppression of tumorigenicity 2 (ST2), which activates basophils, mast cells, eosinophils, group 2 innate lymphoid cells, and CD4+ T cells[7]. The IL-33–ST2 axis is notably prominent in the pathogeneses of several allergic disorders, including asthma, atopic dermatitis, and eosinophilic esophagitis (EoE)[5,8]. A strong genetic association exists between allergy and the IL-33–ST2 axis, as variants in *IL33* and *IL1RL1* (encodes ST2) confer risk for several allergic diseases[9–13]. Thus, the IL-33–ST2 axis has emerged as a primary target for therapeutic modulation in allergy[5].

IL-33 is distinguished from other cytokines by the extensive post-translational modifications that profoundly modulate its ability to activate ST2-expressing cells. Notably, during apoptosis, IL-33 is proteolytically cleaved by caspases 3 and 7 into forms incapable of activating surface ST2[14]. Following acute necrosis, extracellular IL-33 is cleaved into mature forms by the serine proteases derived from neutrophils[15] and mast cells[16] (e.g., elastase and tryptase, respectively), generating highly active forms of IL-33. Additionally, cysteine oxidation of extracellular IL-33 diminishes its ability to active ST2[17]. From these observations, a model is emerging wherein IL-33 is uniquely regulated by post-translational processes. The potency of IL-33 may have necessitated the development of such complex, post-translational regulatory processes to allow fine-tuning.

An unanswered question concerning IL-33 is the functional significance of its unique nuclear localization and chromatin binding[5,18]. Other nuclear alarmins, including HMGB1[19] and IL-1α[20], are considered to be dual function, as they can also act as transcription factors through their ability to bind DNA. IL-33 directly binds to the nucleosome acidic patch composed of the tails of histones H2A and H2B[21], which has important roles in regulating chromatin structure[22]. Several other nucleosome acidic patch-binding proteins act as transcriptional regulators[23], including high mobility group N2 (HMGN2) and latency-associated nuclear antigen (LANA) of the Kaposi sarcoma herpesvirus. The chromatin-binding domain (CBD) of IL-33 has a remarkably high sequence similarity to that of LANA[21] and is conserved across species[21], and IL-33 promotes chromatin compaction[18,21]. Yet, the nuclear function of IL-33 has not been elucidated.

Herein, we aimed to define the functional significance of the nuclear localization and chromatin binding of IL-33 in epithelial cells. We report that chromatin binding regulates IL-33 release and bioactivity. The intranuclear mobility of IL-33 is substantially slow, curtailing its release during necrosis. We show that IL-33 and histones are released as a high-molecular weight complex and together synergistically activate receptor-mediated signaling. Colllectively, we propose that chromatin binding is a post-translational mechanism that regulates the releasability and ST2-mediated bioactivity of IL-33. As such, we propose the paradigm that nuclear localization of cytokines provides a means for fine-tune regulation of cytokine release, availability and activity.

## Results

**Nuclear IL-33 has no impact on global gene expression.** In order to establish the nuclear function of IL-33, we first confirmed the reported[4,18] nuclear localization of IL-33 protein focusing on human allergic inflammation. Using immunofluorescence with two different antibodies directed against IL-33, only nuclear expression was detected in esophageal epithelial cells in biopsies derived from patients with EoE (Supplementary Fig. 1A, B). Similarly, only nuclear IL-33 protein was detected in ex vivo-cultured primary esophageal epithelial cells (Supplementary Fig. 1C) and in an esophageal epithelial cell line (TE-7) engineered to constitutively overexpress IL-33 (Supplementary Fig. 1D). We then overexpressed IL-33 in epithelial cells lacking both endogenous IL-33 and the IL-33 receptor ST2 with the goal of testing the transcriptional effects of nuclear IL-33. In particular, we used lentivirus-mediated, stable transduction to engineer doxycycline (Dox)-inducible overexpression of wild-type (WT) IL-33. Single-cell clones were generated via limiting dilution to ensure uniform expression throughout the population. IL-33 protein expression was efficiently induced with Dox treatment (Fig. 1a, b and Supplementary Figure 7). Genome-wide RNA sequencing (RNA-seq) after 48-h treatment with Dox or vehicle revealed that increased *IL33* reads were detected after Dox in WT IL-33-overexpressing cells but not in control cells nor in vehicle-treated cells (Fig. 1c, d). Differentially expressed genes for each individual clone were identified using loose criteria (RPKM ≥ 1 in at least one sample, Benjamini-Hochberg False Discovery Correction [5%], and a fold change ≥1.5 upon treatment with Dox). Levels of 102 genes were found to change in at least one of the three clones overexpressing WT IL-33 (Fig. 1e); however, *IL33* was the only gene differentially expressed in all three clones (Fig. 1e and Supplementary Table 1). After decreasing the stringency even further by removing the restriction on fold change, *MMP13* was found to be differentially expressed in all three clones overexpressing WT IL-33. However, *MMP13* levels also changed in control cells (Supplementary Table 1), consistent with its known regulation by Dox[24]. Collectively, these results demonstrate that the presence of nuclear IL-33 does not alter a consistent transcriptional program in esophageal epithelial cells under the conditions tested.

**IL-33 tightly and dynamically binds chromatin.** Our unexpected inability to determine a gross effect of IL-33 on the cellular transcription program led us to characterize its biophysical properties. We assessed the strength of the physical association of WT IL-33 with chromatin in esophageal epithelial cells by performing biochemical fractionation with serial extractions of increasing stringency (Fig. 2a). Pools of cells overexpressing WT IL-33, rather than the single-clones examined in Fig. 1, were studied in order to minimize the potential for variable results secondary to clonal selection. Of the three other proteins measured by western blot (the cytoplasmic protein heat shock protein 90, histone H3, and the nuclear matrix protein lamin B), the distribution of IL-33 among the different fractions was most similar to that of histone H3, with detectable amounts of IL-33 and H3 present in the S3, S4, and P4 fractions (Fig. 2b, c and Supplementary Figure 7). Notably, even 2 M NaCl did not fully extract IL-33 from chromatin. Conversely, a truncated form of IL-33 (composed of amino acids 112–270) lacking the chromatin binding domain was fully extracted by Triton X-100 (Fig. 2d, e and Supplementary Figure 7). These data show that WT IL-33,

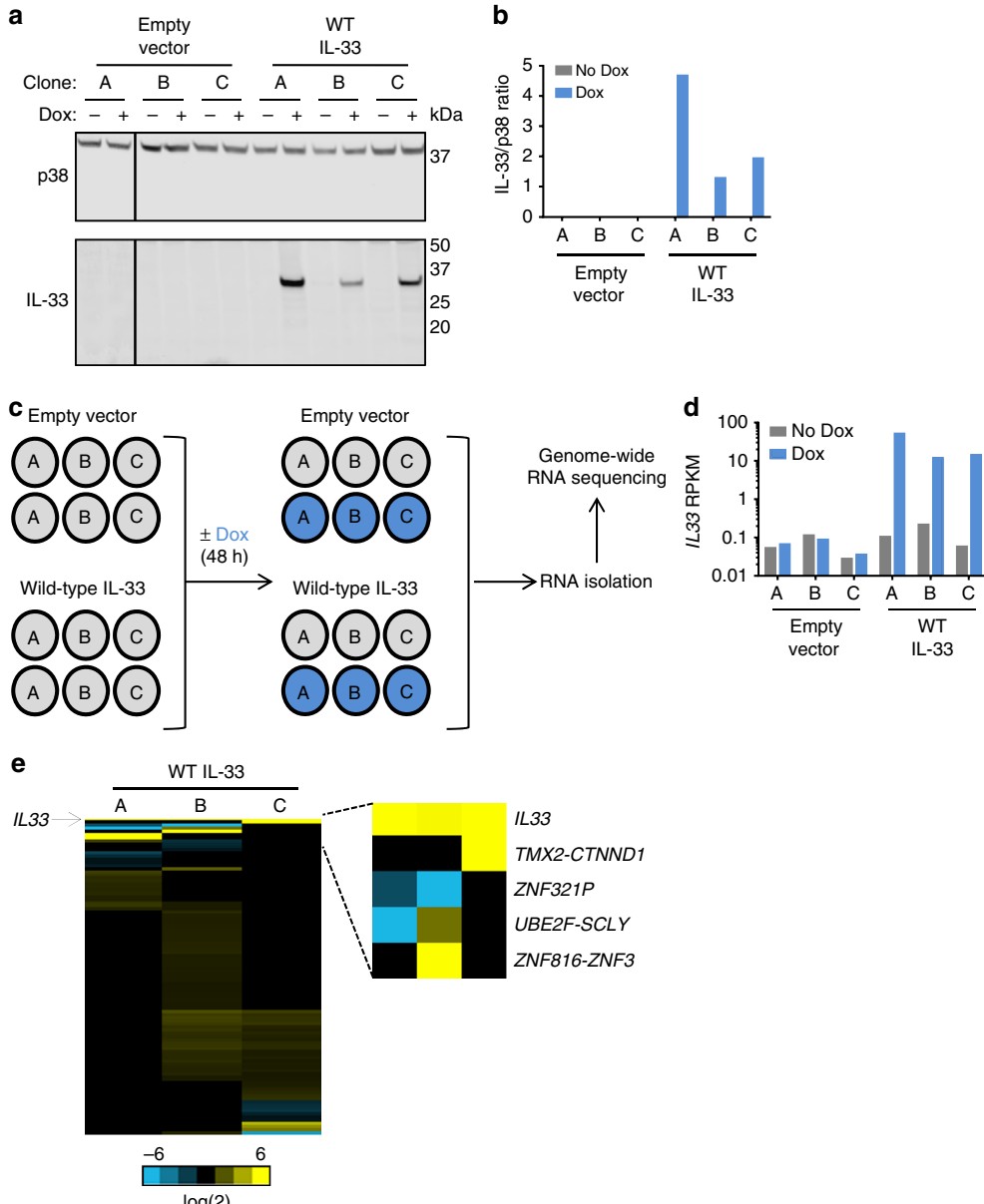

**Fig. 1** Nuclear IL-33 does not impact gene expression. **a** Western blot analysis of IL-33 protein expression, with quantification in **b**, in indicated single-cell clones of TE-7 cells with stable, doxycycline (Dox)-inducible expression of wild-type (WT) IL-33 or the empty vector with or without treatment for 48 h with Dox. Total p38 was used as a loading control. **c** TE-7 single-cell clones from **a** were treated with Dox for 48 h, and then RNA was isolated and subjected to genome-wide RNA sequencing (RNA-seq) analysis. **d** *IL33* RPKM levels in samples subjected to RNA-seq in **c**. **e** Heatmap of fold change in expression in each single-cell clone overexpressing WT IL-33 of genes with statistically significant differential expression in at least one clone. Differentially expressed genes (DEG) were identified by filtering on RPKM ≥ 1, Benjamini–Hochberg false discovery rate (FDR) of 5%, and fold change (FC) ≥ 1.5. Depicted is the $\log_2$ ratio of gene expression in cells treated with Dox compared to cells without Dox. In **b** and **d**, bars represent values of individual samples. IL, interleukin

but not truncated IL-33, tightly binds chromatin in esophageal epithelial cells.

Next, we assessed localization of IL-33 within chromatin with C-terminal green fluorescent protein (GFP) fusion proteins of WT IL-33 or truncated form of IL-33 composed of amino acid residues 112–270 that was unable to bind chromatin[18] (Fig. 3a). Live-cell confocal microscopy showed that WT IL-33–GFP was restricted to the nucleus as expression was not detected outside of the regions containing the DNA-binding dye Hoechst 33342 (Fig. 3b). Moreover, WT IL-33–GFP exhibited a high correlation with Hoechst 33342 within the nucleus (the Pearson coefficient was 0.55 ± 0.02 [mean ± SEM]). There was not a statistically

significant difference between the co-localization of Hoechst 33342 with WT IL-33–GFP vs. H2B-GFP (Holm–Sidak multiple comparisons test $p = 0.59$) (Fig. 3c), indicating that IL-33–GFP is enriched in regions of heterochromatin, consistent with prior findings[18]. In contrast to WT IL-33–GFP, truncated IL-33–GFP, which was engineered to not bind chromatin, was diffusely expressed in the cytoplasm and nucleus. Additionally, there was not a statistically significant difference in the co-localization of Hoechst 33342 with truncated IL-33–GFP vs. GFP alone ($p = 0.49$) within the nucleus. Because this truncated form of IL-33 is missing a significant amount of the protein in addition to the chromatin binding domain, we also generated

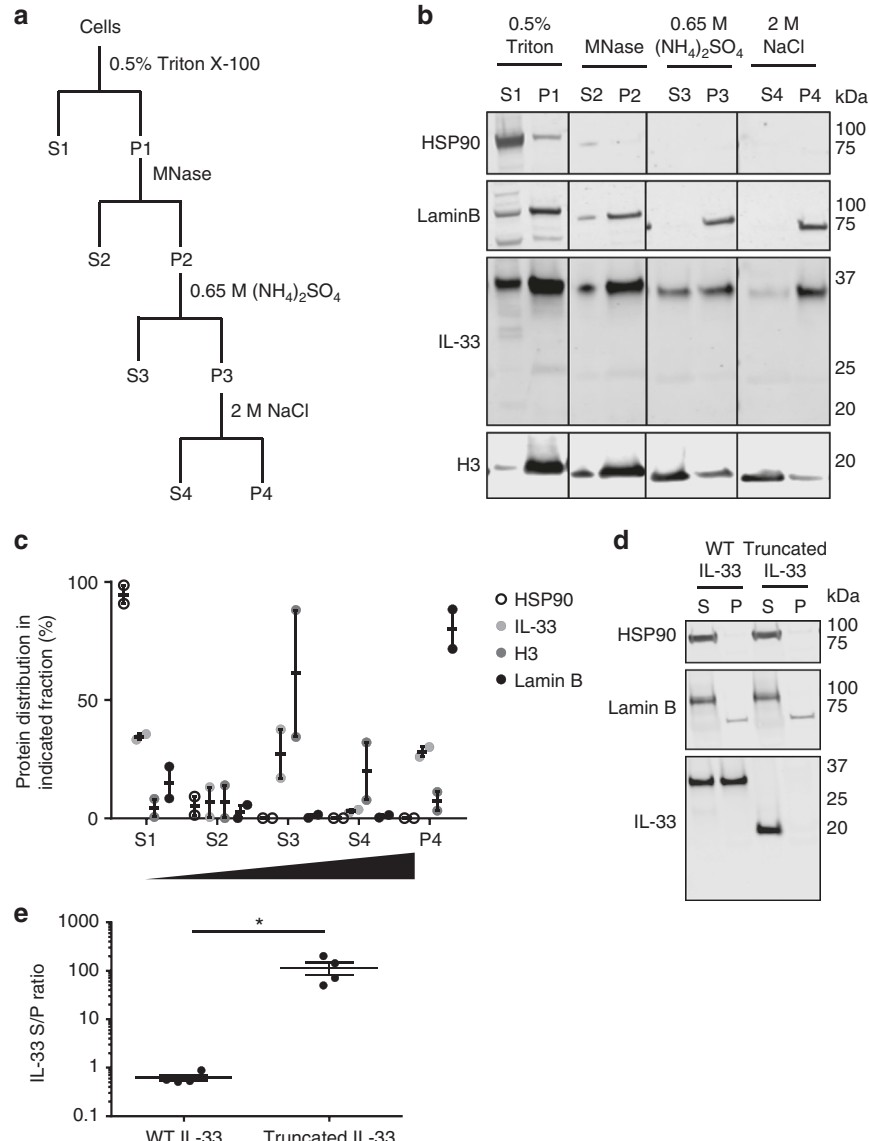

**Fig. 2** IL-33 tightly associates with chromatin. **a** Schematic of biochemical fractionation of IL-33-overexpressing TE-7 cells with serial extraction with indicated treatments. **b** Representative western blot with quantitation of two independent experiments in **c**. S refers to proteins extracted with each serial treatment, and P represents unextracted proteins remaining in the residual pellet. Graph in **c** depicts the percentage of a protein's distribution in an indicated fraction compared to the total (S1+S2+S3+S4+P4). **d**, **e** Pools of TE-7 cells with Dox-inducible overexpression of wild-type or truncated IL-33 were treated with Triton X-100 (0.5%). **d** is a representative western blot, and **e** shows quantification of the ratio of IL-33 detected in the supernatant to pellet (S/P) after Triton X-100 treatment from three independent experiments. Depicted is mean ± standard error of the mean. *, $p < 0.05$ using parametric two-tailed Student's $t$-test. H3: histone H3, HSP90: heat shock protein 90, MNase: micrococcal nuclease, WT: wild-type

IL-33–GFP-fusion proteins (R48A and 47AAA49) with mutations within the chromatin binding domain (Supplementary Fig. 2A) that diminish binding to chromatin by specifically altering docking to the H2A-H2B dimer[21]. The strength of chromatin binding of these mutant IL-33–GFP-fusion proteins was intermediate compared with WT and truncated IL-33 as assessed by western blot analysis of 0.5% Triton-X 100 cell lysates (Supplementary Fig. 2B, C and Supplementary Figure 9). Importantly, the correlation of Hoechst 33342 and R48A IL-33–GFP or 47AAA49 IL-33–GFP was less than WT IL-33–GFP (Supplementary Fig. 2D, E). These results indicate that the nuclear restriction and heterochromatin enrichment of IL-33–GFP occur because of chromatin binding. Finally, as a control, the Pearson coefficient within the nucleus of Hoechst 33342 with mouse IL-1α–GFP was lower than that of WT IL-33–GFP ($p < 0.0001$) but indistinguishable from GFP alone

($p = 0.59$). The mouse version of IL-1α was examined as this construct has been independently verified by others[25]. This finding indicates that IL-1α–GFP is not enriched in heterochromatin despite the fact that it is in the same cytokine family as IL-33[3]. In summary, these data demonstrate that chromatin binding causes IL-33 to be enriched in regions of heterochromatin in epithelial cells.

We next quantitated the kinetics of the interactions of IL-33 with chromatin using fluorescence recovery after photobleaching (FRAP). WT IL-33–GFP exhibited a relatively slow intranuclear mobility, with recovery of 50 and 70% of the fluorescence within the ROI taking 13 ± 1 and 30 ± 2 s (mean ± SEM), respectively (Fig. 4a, b and Supplementary Table 2). The intranuclear mobility of truncated IL-33–GFP was indistinguishable from GFP alone. In independent experiments, the intranuclear mobilities of R48A IL-33–GFP and 47AAA49 IL-33–GFP were greater than

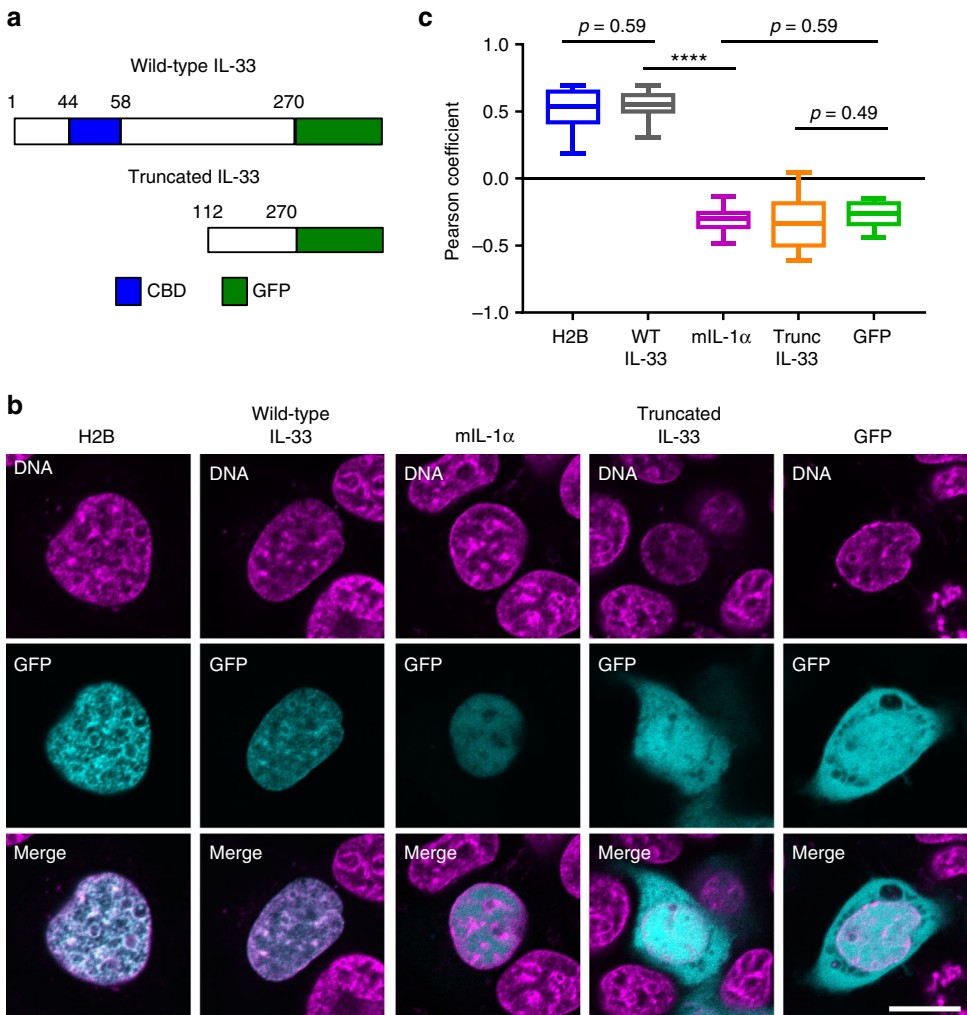

**Fig. 3** IL-33 exhibits heterochromatic localization. **a** Schematic of wild-type and truncated IL-33–GFP-fusion proteins. **b** Representative live-cell images from one experiment from **c**. Top row indicates Hoechst 33342 DNA dye (magenta), middle row indicates presence of GFP-fusion protein (turquoise), and bottom row contains merged images. Scale bar is 10 μm. **c** Quantification of the Pearson correlation coefficient within the nucleus between fluorescence from Hoechst 33342 DNA dye and GFP after transient transfection of plasmid encoding indicated GFP-fusion protein into TE-7 cells. Graph is a box-and-whiskers plot with center line indicating median value, box limits showing upper and lower quartiles, and whiskers indicating the minimum and maximum values. Data are combined from 3–5 independent experiments (between 13 and 29 cells each). One-way ANOVA $p$ value was <0.0001, and depicted are $p$ values from Holm–Sidak multiple comparisons test. ANOVA: analysis of variance, CBD: chromatin binding domain, GFP: green fluorescent protein, H2B: histone H2B, IL: interleukin, mIL-1α: mouse IL-1α, Trunc: truncated, WT: wild-type

WT IL-33–GFP (Supplementary Fig. 3A, B and Supplementary Table 3). These results indicate that the slow mobility of WT IL-33–GFP was dependent on chromatin binding. Notably, WT IL-33–GFP was dramatically less mobile than IL-1α–GFP, which had a shorter 70% recovery (3.5 ± 0.2 vs. 29.7 ± 2.3 s, respectively, $p <$ 0.0001). In fact, the 50% recovery of IL-1α–GFP was too fast to measure, as previously reported[25]. However, WT IL-33–GFP was dramatically more mobile than the core histone H2B-GFP, which had a 50% recovery time beyond the time of measurement of the experiment. Furthermore, WT IL-33–GFP exhibited a smaller immobile fraction (interpreted as the proportion of the protein that does not change location over the course of the experiment) than H2B-GFP. Overall, these data demonstrate that IL-33 exhibits dynamic binding to chromatin with a much higher average residence time than IL-1α.

**Chromatin binding reduces IL-33 release during necrosis**. We hypothesized that the tight dynamic binding of IL-33 to chromatin could be a mechanism of regulating the extracellular release of IL-33. To test this hypothesis, we compared the extracellular release of IL-33 variants after inducing necrosis with a 4-h treatment with calcium ionophore A23187 (20 μM). Calcium ionophore treatment increased the release of truncated IL-33 compared to WT IL-33 (supernatant/pellet ratio of 1.4 ± 0.5 vs. 0.2 ± 0.1 [mean ± SEM], respectively; two-way ANOVA interaction term $p = 0.05$), with a 7.8-fold change between truncated and WT IL-33 (Fig. 5a, b and Supplementary Figure 8). Furthermore, WT IL-33 was highly cell-associated despite marked necrosis, indicating intracellular retention; a similar rate of necrosis was observed between cells overexpressing truncated and WT IL-33 ($p > 0.90$) (Fig. 5c). In independent experiments, we assessed the release of truncated and WT IL-33 in response to cryoshock. Accordingly, cryoshock induced increased release of truncated IL-33 compared to WT IL-33 (supernatant/pellet ratio of 7.4 ± 0.7 vs. 1.1 ± 0.3, respectively; Student's $t$-test $p < 0.001$), with a 7.0-fold change between truncated and WT IL-33 (Fig. 5d, e and Supplementary Figure 8); there were no statistically significant differences in cellular viability between WT

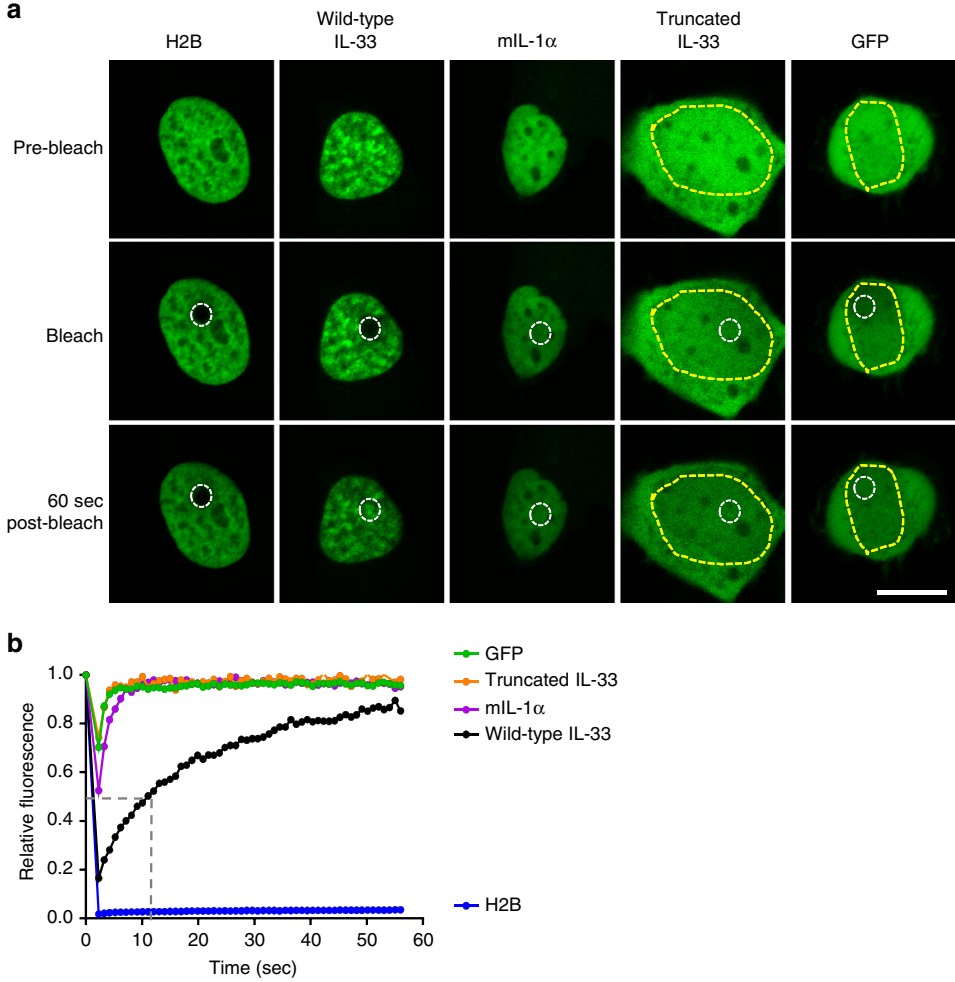

**Fig. 4** IL-33 dynamically binds to chromatin. **a** Fluorescence recovery after photobleaching (FRAP) was performed on TE-7 cells after transient transfection with plasmid encoding the indicated GFP-fusion proteins. A region of interest (ROI) (white dashed lines) was bleached, and fluorescence within that ROI was determined continuously over the following 60 s. Representative images of GFP fluorescence (green) from before bleach (top row), immediately after bleach (middle row), and 60 s post-bleach (bottom row) are shown for the indicated GFP-fusion protein. For GFP-fusion proteins with significant cytoplasmic localization, the nucleus is outlined with yellow dashed lines. Scale bar is 10 μm. **b** Representative FRAP experiment showing fluorescence within ROI during bleach and for 60 s post-bleach for each GFP-fusion protein. Dashed gray lines indicate 50% recovery for wild-type IL-33–GFP. IL interleukin, mIL-1α mouse IL-1α

and truncated IL-33-expressing cells ($p > 0.80$) (Fig. 5f). In independent experiments, the supernatant/pellet ratio of R48A IL-33–GFP and 47AAA49-IL-33-GFP was higher than WT IL-33–GFP (Supplementary Fig. 4A, B and Supplementary Figure 9). These results demonstrate that chromatin binding decreases the extracellular release and increases the intracellular retention of IL-33 after necrosis.

We next visualized the kinetic release of IL-33 from necrotic cells using live-cell, time-lapse confocal microscopy. Following impaired membrane integrity, WT IL-33–GFP exhibited increased intracellular retention compared to IL-1α–GFP, truncated IL-33–GFP, and GFP alone (Fig. 6a, b and Table 1). The retention of WT IL-33–GFP was consistently lower than that of H2B-GFP. In independent experiments, R48A IL-33–GFP and 47AAA49 IL-33–GFP exhibited decreased intracellular retention compared with WT IL-33–GFP (Supplementary Fig. 5A, B and Supplementary Table 4). Notably, a slow, steady decrease in intracellular levels over time was observed for WT IL-33–GFP but not H2B-GFP. Both WT and truncated IL-33 were detected by western blot analysis of supernatants after treatment with Triton X-100 in independent experiments (Fig. 2d, e and Supplementary Figure 7), indicating that their differential releasability was not

reflective of differential degradation. Collectively, these data indicate that IL-33 exhibits a relatively slow, durable extracellular release upon loss of membrane integrity.

**Released chromatin-bound IL-33 synergistically activates ST2.** To test whether IL-33–chromatin complexes were released during necrosis, we subjected necrotic supernatants to size-exclusion chromatography. Approximately 70% of WT IL-33 was detected as high molecular weight (HMW) complexes (e.g., the void volume of the column) (Fig. 7a, b). The fractions with the highest concentrations of WT IL-33 (Fig. 7a) were from fractions that contained molecular species of at least 100 kDa. In contrast, truncated IL-33 was predominantly detected as a low molecular weight (LMW) species (Fig. 7a, b). Western blot analysis of pooled fractions demonstrated that WT IL-33 was predominantly present as HMW species, whereas truncated IL-33 was not (Fig. 7c and Supplementary Figure 8). In addition, DNA and histones were readily detected in the HMW fractions, whereas they were undetectable in LMW fractions (Fig. 7c, d and Supplementary Figure 8). Furthermore, a detectable amount of released WT IL-33 co-immunoprecipitated with endogenous

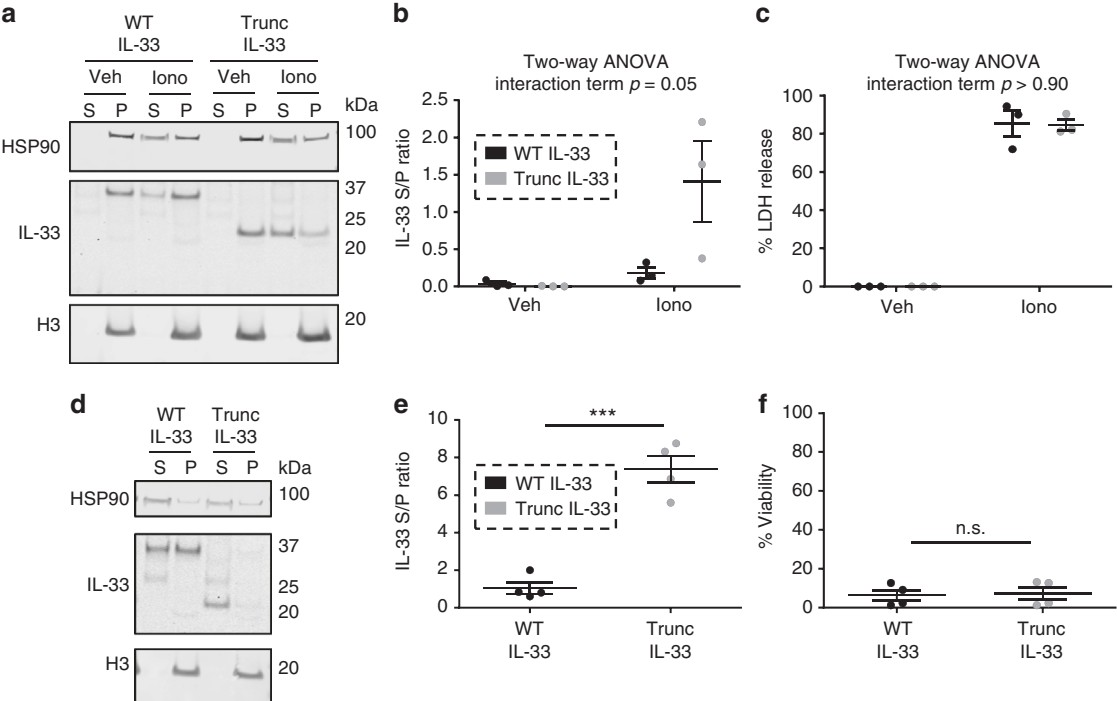

**Fig. 5** Chromatin binding reduces IL-33 release during necrosis. **a–c** TE-7 pools with stable, Dox-inducible overexpression of wild-type (WT) or truncated (Trunc) IL-33 were treated for 4 h with 20 μM of calcium ionophore A23187 (Iono) or vehicle (Veh). **a** shows a representative western blot. **b** is quantification of the ratio of IL-33 detected in the supernatant vs. the pellet (S/P) by western blot. **c** is quantification of the percent release of lactate dehydrogenase (LDH) using enzymatic activity assay, defined as LDH activity in supernatant divided by the total amount in the supernatant plus pellet [S/(S+P)]. **b**, **c** depict the mean and standard error of the mean of three independent experiments. Two-way ANOVA interaction terms for IL-33 and % LDH release were $p = 0.05$ and $p > 0.90$, respectively. **d–f** The same pools of IL-33–overexpressing TE-7 cells were subjected to cryoshock. **d** shows a representative western blot. **e**, **f** are quantification of the ratio of IL-33 detected in supernatant to pellet (**e**) or cell viability as assessed by Trypan blue exclusion (**f**); presented is the mean and standard error of the mean of three independent experiments. \*\*\*, $p < 0.001$ and n.s.: not significant by Student's $t$-test. ANOVA: analysis of variance, Dox: doxycycline, H3: histone H3, HSP90: heat shock protein 90, IL: interleukin, P: cell pellets, S: supernatants

histone H2B using an anti-H2B antibody following cryoshock-induced cellular necrosis (Fig. 7e and Supplementary Figure 8). Finally, we assessed whether IL-33 retains the ability to activate the ST2 receptor when in complex with histones by assessing cellular activation of HMC-1 mast cells, a common biosensor for ST2 activity[26,27], after treatment of supernatants from cells overexpressing WT IL-33 generated via cryoshock after concentration using a centrifugal filter with a 100 kDa exclusion. IL-8 production was detected after treatment with either the input or the concentrate in a dose-dependent fashion and was diminished with pre-treatment with an ST2-neutralizing antibody to a greater degree with higher dilution (Fig. 7f). These results demonstrate that IL-33 is released in complex with chromatin during necrosis.

We next tested whether histones and IL-33 cooperatively interacted through ST2 by assessing cellular activation of HMC-1 mast cells following treatment with acid-extracted histones (Fig. 8a) and recombinant WT IL-33. We first confirmed that purified histones had negligible amounts of DNA remaining after acid extraction (Supplementary Fig. 6A) and that recombinant WT IL-33 interacted directly with acid-purified histones by co-immunoprecipitation (Supplementary Fig. 6B). Next, cells were treated with WT IL-33 in the presence and absence of acid-extracted histones and the supernatant was assessed for the release of an array of cytokines. Notably, there was synergistic release of 12 of the 42 cytokines assessed following co-treatment of WT IL-33 and histones (Fig. 8b, Supplementary Tables 5, 6), ranging from 1.5 to 98-fold compared to IL-33 treatment alone (Supplementary Table 6). The most markedly induced cytokine was CCL2, which was induced 8 and 822-fold by IL-33 in the

absence and presence of histones, respectively. Notably, treatment with histones alone had no effect on CCL2 secretion, although it had relatively modest effects on other cytokines (Fig. 8b, Supplementary Tables 5, 6). IL-8 secretion was induced 17 and 84-fold in the absence and presence of histones, respectively. This synergistic induction of IL-8 was reproduced using an IL-8 specific enzyme-linked immunosorbent assay (ELISA) rather than the multiplex array (Fig. 8c) and was abolished with pre-treatment with an ST2-neutralizing antibody but not the control IgG (Fig. 8c). The cooperative activity was detectable across a broad range of doses of acid-purified histones (Supplementary Fig. 6C) and WT IL-33 (Fig. 8d, g). Additionally, cooperative activity was observed with purified recombinant histones (Supplementary Fig. 6D), ruling out a requirement for any residual DNA or non-histone proteins present in the acid-extractions. As a control, no cooperativity was seen between acid-purified histones and truncated IL-33 (Fig. 8e, h) or IL-1α (Fig. 8f, i), consistent with IL-1α binding directly to DNA rather than histones[2]. In independent experiments, cellular signaling induced by WT IL-33 was increased by histones, as assessed by phosphorylation of NFκB (Fig. 8j, k and Supplementary Figure 8), a principal transcription factor downstream of ST2[6]. In total, these data demonstrate that IL-33 and histones synergize to induce ST2-dependent signaling and cytokine production.

## Discussion

The cytokine IL-33 is a potent extracellular activator of innate immunity; however, the reasons for its unique nuclear

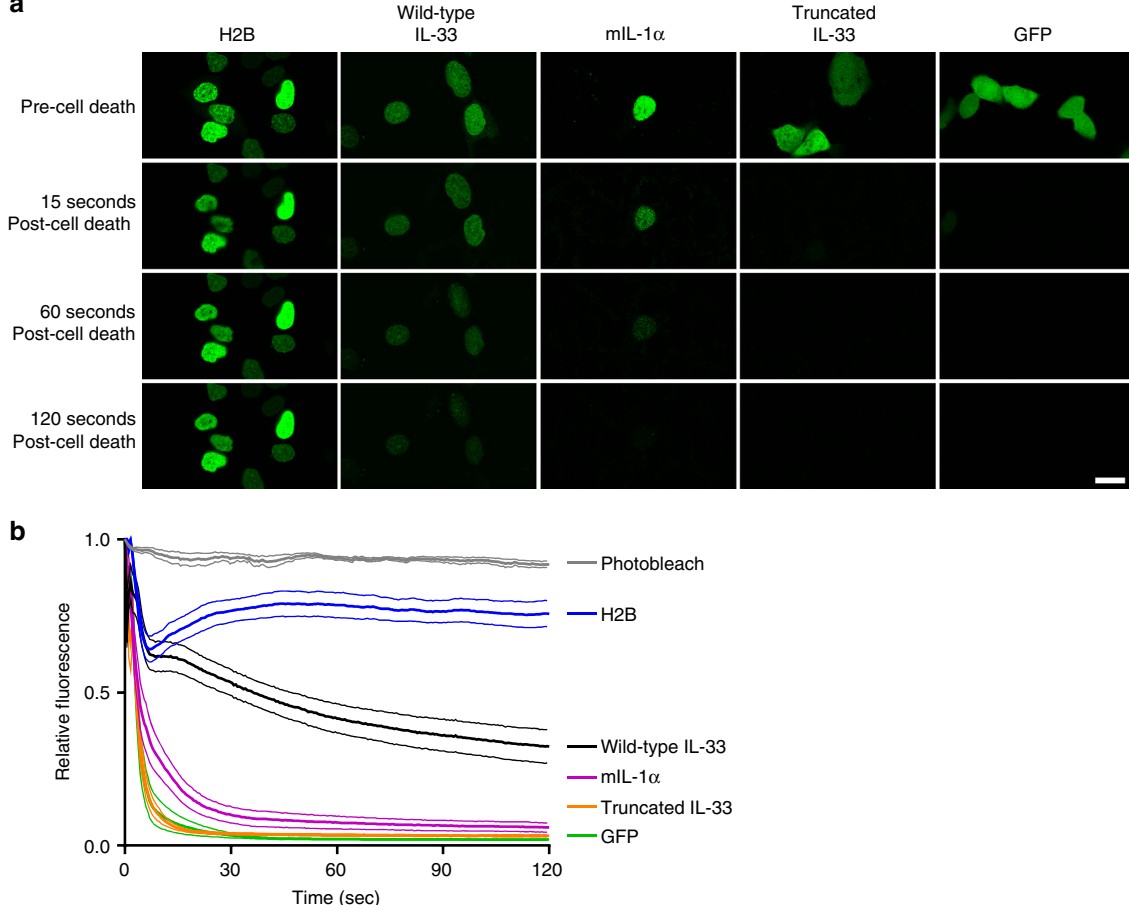

**Fig. 6** Chromatin-binding dynamics regulate IL-33 release kinetics. **a** TE-7 cells were transiently transfected with plasmid encoding the indicated GFP-fusion protein and then treated with Triton X-100 (0.13%) to induce necrosis. Depicted are images from a representative experiment showing 5 s prior to cell killing by the Triton X-100 ("Pre-Cell Death"; top row) and 15, 60, and 120 s after death ("Post-Cell Death"; second, third, and fourth rows, respectively) from a representative experiment. Scale bar is 10 μm. **b** Quantification of intracellular fluorescence intensity from three combined independent experiments. Data are normalized to time zero, which is when Triton X-100 was added. Photobleach curve was generated in the absence of Triton X-100. For each depicted curve, the central, thicker line indicates the mean and the bordering, thinner lines of the same color indicate the standard error of the mean. GFP: green fluorescent protein, IL: interleukin, mIL-1α: mouse IL-1α

| Table 1 Quantitation of kinetics of release | | | |
| --- | --- | --- | --- |
| **Protein** | **15 s** | **60 s** | **120 s** |
| GFP | 6.9 ± 0.8% | 2.1 ± 0.1% | 1.9 ± 0.1% |
| Truncated IL-33 | 5.6 ± 0.2% | 3.5 ± 0.1% | 3.2 ± 0.1% |
| IL-1α | 19.3 ± 1.3% | 7.5 ± 0.6% | 5.8 ± 0.5% |
| Wild-type IL-33 | 60.9 ± 1.3% | 41.5 ± 1.3% | 32.3 ± 1.5% |
| Histone H2B | 70.3 ± 1.4% | 78.6 ± 1.5% | 75.7 ± 1.5% |

Mean and standard error of the mean of the percentage of intracellular fluorescence intensity at the indicated timepoints from Fig. 6 relative to time zero
*GFP* green fluorescent protein, *IL* interleukin, *s* second

localization and chromatin binding have remained enigmatic. Herein, we have elucidated the biophysical properties of IL-33 binding to chromatin and uncovered a functional role of this process. In particular, we have made the observations that (1) IL-33 retains its predominantly nuclear localization in vivo, even under conditions classically associated with its release, such as human allergic inflammation; (2) despite its nuclear localization, IL-33 does not affect homeostatic gene expression as assessed by genome-wide transcript profiling of esophageal epithelial cells engineered to overexpress IL-33; (3) the chromatin-binding dynamics of IL-33 are remarkably slower than IL-1α[25], typical DNA-binding transcription factors[28], and previously studied mammalian nucleosome acidic patch-binding proteins;[29,30] (4) IL-33 chromatin binding curtails its extracellular release with relatively high nuclear retention even within necrotic cells as assessed by analysis of IL-33 structural variants; (5) upon disruption of membrane integrity, there is a relatively slow release of IL-33 over time; (6) IL-33 is released extracellularly in complex with chromatin; and (7) IL-33 synergizes with histones to induce ST2-dependent signaling and release of soluble mediators relevant to allergic inflammation. On the basis of these findings, we propose that the chromatin-binding properties of IL-33 have emerged to decrease its bioavailability for release while simultaneously enhancing both the duration of its release and ST2-related bioactivity (see model in Fig. 9). Thus, we have uncovered the functional role for the nuclear localization of IL-33 by demonstrating that its high-affinity binding to chromatin is an additional post-translational mechanism to regulate its activity. We speculate that the high degree of unique post-translational regulation of IL-33, which we now attribute in part to its chromatin binding, likely arose due to the relative potency of extracellular IL-33. For example, focusing solely on eosinophils, IL-33 induces expression of >10-fold more genes than the classic pro-inflammatory cytokine IL-4, which uses a classical secretion mechanism[7].

We find no evidence in our study for a role for nuclear IL-33 in regulating gene expression, consistent with a recent, elegant, proteomic analysis that found no reproducible effect of nuclear IL-33 on protein expression in primary endothelial cells[31]. It is notable that we obtained similar results with an alternate cell type, namely epithelial cells. In addition, we used an independent approach by inducing overexpression of IL-33, whereas the prior work focused on a gene-silencing approach. The inducible system likely minimizes compensatory mechanisms that could potentially occur with extended exposure to IL-33. Furthermore, the cells used herein did not express ST2 nor respond to extracellular IL-33. Nevertheless, we cannot exclude the existence of context-dependent intracellular functions of nuclear IL-33.

Our finding that loss of IL-33 chromatin binding increases its extracellular release by mediating intracellular retention is strengthened by the use of multiple independent necrotic stimuli and IL-33 structural variants. Notably, the changes in IL-33 extracellular release and biochemical properties, including intra-nuclear mobility, that were identified with the truncated form of IL-33 were reproduced with targeted mutations within the chromatin binding domain (R48A and 47AAA49) that are known to specifically diminish the interaction of IL-33 with the nucleosome acidic patch[21]. Collectively, these findings build upon a previous elegant study showing that deletion of the chromatin binding domain of IL-33 by an endogenous knock-in approach causes overwhelming lethal eosinophilic inflammation not seen in control mice[32].

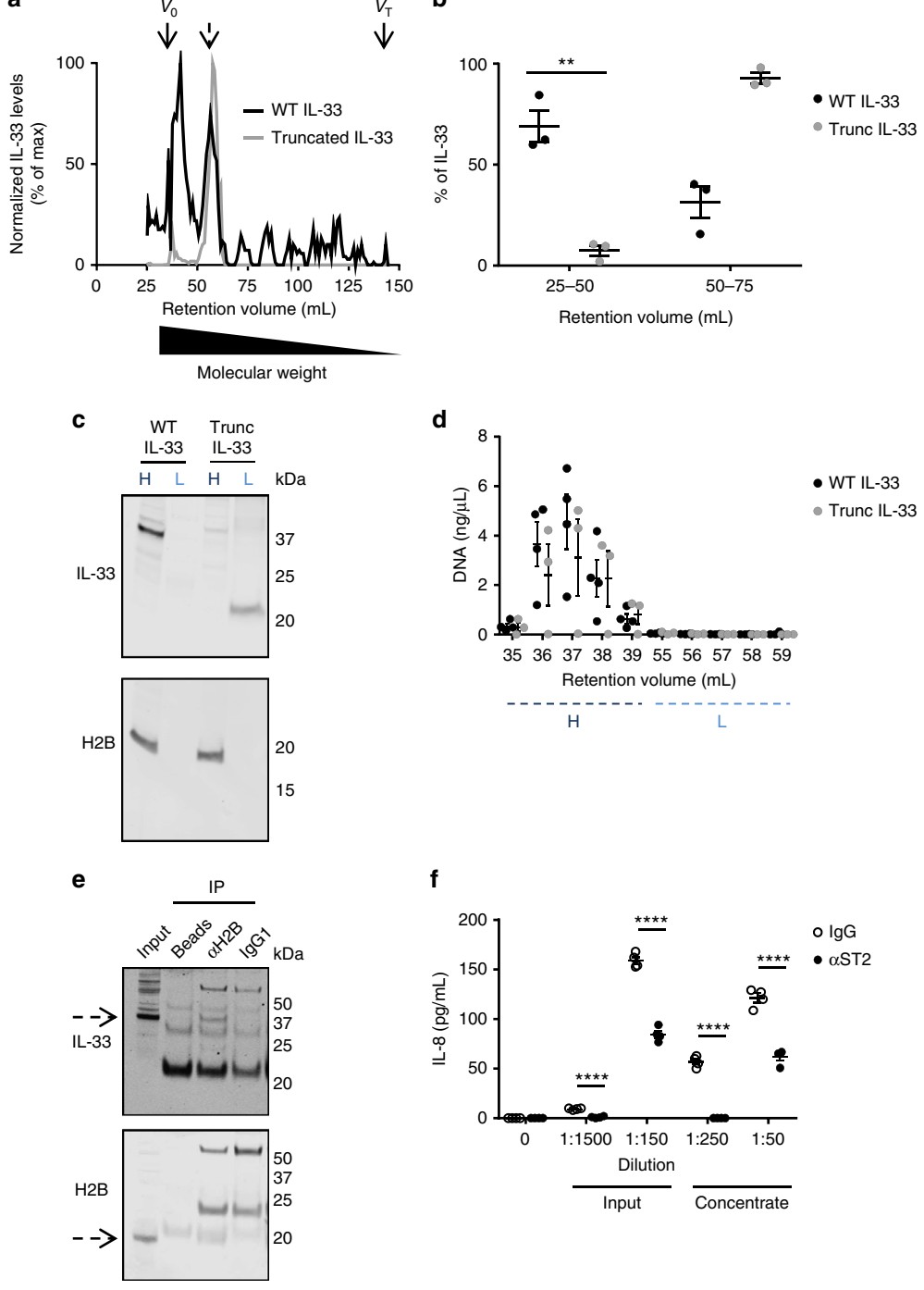

We observed synergy between IL-33 and histones in the ST2 bioassay using a broad range of doses of both histones (1.7 to 26.6 µg/mL) and recombinant WT IL-33 (1.5–46.7 nM). To the best of our knowledge, there have been no studies examining the concentrations of extracellular histones in the microenvironment surrounding epithelial cells, and limited studies have examined plasma histone levels. The median plasma nucleosome concentration in hepatocellular carcinoma patients was found to be approximately 1.5 µg/mL with a max of 6.4 µg/mL[33]. Additionally, in vitro studies of the effects of extracellular histones have examined histone concentrations greater than those used herein[34–36]. These collective data suggest that the amounts of histones examined in the ST2 bioassays may be physiologically relevant.

Several potential molecular mechanisms could underlie the observed synergy between IL-33 and histones. Histones could enhance the binding of IL-33 to its receptor, ST2. Conversely, there could be synergistic downstream signaling secondary to activation of other cell surface receptors by histones. Consistent with this idea, histones have been proposed to act as damage-associated molecular patterns to promote inflammation through activation of Toll-like receptors, such as TLR2, TLR4, and TLR9[37–39]. Importantly, we did not observe any effect of histones alone on cellular signaling in the ST2 biosensor cell. Future studies should investigate the molecular mechanisms, as well as determine whether co-treatment of ST2-expressing cells with IL-33 and histones causes different functional properties, such as cellular activation and cytokine release, than either alone. This is likely to occur in light of our findings that co-treatment of mast cells with histones and IL-33 results in increased release of pro-inflammatory cytokines (IL-6[40]) and known chemoattractants of neutrophils (IL-8[40]), eosinophils (CCL3[41]), activated T cells (CCL2[41]), and monocytes (CCL2, CCL3[41]).

Full-length IL-33 can be proteolytically processed into mature forms with a greater ability to activate the ST2 receptor[15,16]. In the ST2 bioassay experiments, co-treatment with acid-purified histones did not increase the ST2 bioactivity of recombinant WT IL-33 to the level of truncated IL-33. This would make it appear that processed mature forms would still have more ST2 bioactivity in vivo. There are limitations to this interpretation, as the recombinant proteins were generated by different procedures (in vitro wheat germ vs E. coli). Moreover, the recombinant WT IL-33 was GST-tagged whereas the truncated IL-33 was not. Future experiments controlling for differences in protein production will be required to more accurately compare the potency of histone-bound WT IL-33 and enzymatically processed mature forms.

IL-33 is often grouped together with thymic stromal lympho-poietin (TSLP) and IL-25 as innate, epithelium-derived cytokines expressed at mucosal surfaces that promote type 2 immune responses[42]. However, IL-33 is the only one of the three that is stored in the nucleus as a pre-formed molecule bound to chromatin. Its chromatin-binding properties distinguish IL-33 from TSLP, IL-25, and other prototypical alarmins such as IL-1α and HMGB1. IL-33 has much slower dynamics of chromatin binding than either IL-1α or HMGB1[43,44]. HMGB1 is known not to be released in similar complexes during necrosis[45], and no IL-1α–chromatin complexes have been described under necrotic conditions. IL-1α also was not found to have cooperative downstream receptor signaling with purified histones, likely due to the fact that IL-1α does not directly interact with histones like IL-33 does. In addition, IL-33's chromatin-binding properties also separate it from other nucleosome acidic patch-binding proteins. To the best of our knowledge, no mammalian nucleosome acidic patch-binding proteins have been shown to exhibit an intranuclear mobility as low as that of IL-33 as assessed by FRAP. Other nucleosome acidic patch-binding proteins, such as HMGN1[29] and HMGN2[30], are intracellular transcriptional regulators with minor extracellular immune-related activities[46]. These findings fit with our proposed model that the unique properties of IL-33 chromatin binding arose to fine-tune its extracellular activity.

IL-33 is classically associated with promoting allergic responses through extracellular release during necrosis. However, in assessing the localization of IL-33 in human allergic esophageal tissue, we only detected nuclear expression. This is consistent with previous studies that have only detected IL-33 nuclear localization in the bronchial epithelium, as detected by immunostaining, despite high levels of IL-33 present in the bronchoalveolar lavage fluid[47]. Additionally, in other tissues[48,49] IL-33 is localized to the nucleus. Perhaps only a small fraction of IL-33 is secreted, which remains relatively undetectable due to its low level, clearance mechanisms, or cysteine oxidation[17], which may reduce its immunodetectability. Indeed, the biochemical fractionation studies clearly demonstrate that IL-33 has very tight binding to chromatin as high-salt extractions did not completely release it. Additionally, extracellular IL-33 may have a very short half-life due to its binding to the decoy soluble ST2, internalization upon binding cell-surface ST2, and proteolytic degradation[50,51]. Finally, patients with EoE may not be undergoing sufficient cellular stimulation to induce cellular necrosis at the time of the biopsy procurement. It is notable that patients are fasting prior to endoscopy and hence not exposed to the causative food antigens immediately prior to the biopsy collection.

**Fig. 7** IL-33 is released in complex with histones. TE-7 pools with stable, Dox-inducible overexpression of wild-type (WT) or truncated (Trunc) IL-33 were subjected to cryoshock, and the presence of high molecular weight IL-33 species in the supernatants was subsequently determined by size-exclusion chromatography. **a** IL-33 protein levels in different fractions as determined by ELISA, with normalization to the fraction with the highest amount of IL-33. Dashed arrow indicates where WT IL-33 is predicted to elute based off of a previously generated protein standard curve. **b** Proportion of IL-33 present in indicated fractions from **a**. **c** High molecular weight (corresponding to 35–39 mL retention volume, H) and low molecular weight (corresponding to 55–59 mL retention volumes, L) were pooled, concentrated by acetone precipitation, and subjected to western blot analysis for IL-33 and histone H2B. **d** DNA concentration in indicated fractions (high molecular weight [H], low molecular weight [L]) as determined by Qubit. **a**, **c**, **e** depict a representative example of three independent experiments. **b**, **d** depict mean and standard error of the mean of cumulative data from three independent experiments. **e** TE-7 pools with stable, Dox-inducible overexpression of WT IL-33 were subjected to cryoshock, and then co-immunoprecipitation with anti-histone H2B antibody (αH2B), isotype control (IgG1), or Protein A/G beads alone (Beads) was performed. Protein expression in eluates of IL-33 and histone H2B was assessed by western blot analysis. Black dashed arrows indicate bands corresponding to IL-33 and H2B. **f** Supernatants from pools of TE-7 cells with stable, Dox-inducible overexpression of WT IL-33 subjected to cryoshock were concentrated with a centrifugal filter with a 100 kDa molecular weight exclusion. HMC-1 mast cells that had been pre-treated with anti-ST2 or control IgG for 1 h were then treated with dilutions of either the input or concentrate as indicated. Depicted is mean and standard error of the mean of a representative example of two independent experiments. **, $p < 0.01$ by Student's t-test; ****$p < 0.0001$ by Student's *t*-test with Holm–Sidak correction for multiple comparisons. Dox: doxycycline, ELISA: enzyme-linked immunosorbent assay, Ig: immunoglobulin, IL: interleukin, ST2: suppressor of tumorigenicity 2, $V_O$: void volume of column, $V_T$: total volume of column

 

In summary, we have demonstrated functional roles for the chromatin binding of IL-33 in epithelial cells. In particular, we have determined that the chromatin-binding dynamics of IL-33 are remarkably slower than IL-1α[25], typical DNA-binding transcription factors[28], and previously studied mammalian nucleosome acidic patch-binding proteins[29,30]. Furthermore, we have shown that IL-33–chromatin binding curtails IL-33's extracellular release, with relatively high nuclear retention of IL-33 even within

necrotic cells, and promotes a relatively slow release of IL-33 over time. In addition, we have shown that IL-33 is released extracellularly in complex with chromatin and that histones enhance IL-33–induced ST2-dependent signaling. Thus, we have elucidated the functional significance of IL-33 binding to chromatin, showing that chromatin binding regulates both the availability of IL-33 for release and its extracellular cytokine activity. Collectively, we have identified chromatin binding as a

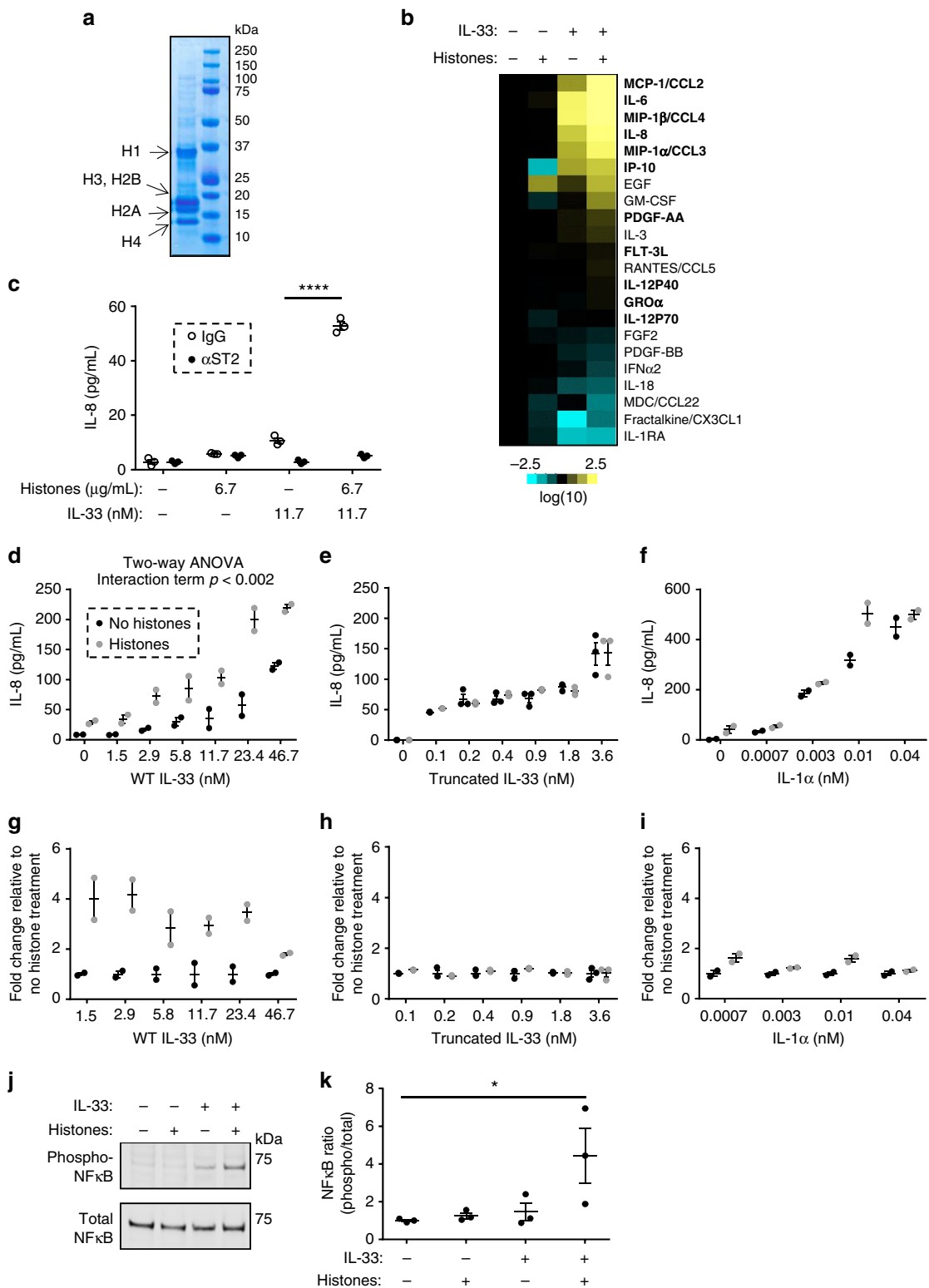

post-translational mechanism that regulates IL-33 release and bioactivity.

## Methods

**Cell culture.** The esophageal epithelial cell line TE-7 (a kind gift of Dr. Hainault, France), 293T human embryonic kidney cell line (purchased from Clontech), and HMC-1 mast cell line were cultured in RPMI-1640 medium (Invitrogen, Carlsbad, CA) supplemented with 5% fetal bovine serum (FBS). Primary esophageal epithelial cells were cultured in keratinocyte serum-free media (KSFM) (Life Technologies, Grand Island, NY) as described previously[52]. All cell lines were tested for mycoplasma contamination before use. The TE-7 cell line is listed in the ICLAC database of commonly misidentified cell lines. The cell line was authenticated by short tandem repeat DNA profiling before use. Additionally, use of this cell line is justifiable as the interaction with chromatin is a fundamental property of IL-33 that has been demonstrated in a wide variety of diverse cell types.

**Generation of plasmids.** In order to generate cells with stable, constitutive overexpression of IL-33, the cDNA for the *Homo sapiens* full-length *IL33* mRNA (NM_001314044.1) was cloned into the pLVX lentiviral vector using the Infusion method (Clontech, CA). In order to generate plasmids encoding proteins with GFP fused to the C-terminus, the Infusion method (Clontech) was used to insert into the peGFP-N1 vector the cDNA for *Mus musculus* full-length *Il1a* (NM_010554.4) or the *Homo sapiens* full-length *IL33* (NM_001314044.1) or a truncated form of *IL33* that only encodes amino acids 112–270, both with a C-terminal FLAG tag (encoding the amino acid residues DYKDDDDK). Plasmid #16680 encoding human H2B-GFP was purchased from Addgene. In order to generate TE-7 cells with stable, doxycycline (Dox)-inducible overexpression of IL-33, the cDNA for the *Homo sapiens* full-length *IL33* or a truncated form of the gene that only encodes amino acids 112–270, both with an additional C-terminal FLAG tag (encoding the amino acid residues DYKDDDDK), were cloned into the pINDUCER20 vector via the Gateway cloning system.

In order to generate plasmids encoding full-length IL-33 with R48A or 47AAA49 mutations, PCR-mediated mutagenesis was performed using pEGFP-IL33-FL or pINDUCER20-IL33-FL as a template to generate a DNA product inclusive of a restriction fragment (HindIII—BbsI for pEGFP-IL33-FL or BsmBI—BstEII for pINDUCER20-IL33-FL) containing one of the following sets of mutations: (1) c.142C>G; c.143G>C (p.Arg48Ala) or (2) c.139C>G; c.140T>C; c.141C>G; c.142C>G; c.143G>C; c.145T>G; C147T>G (p.Leu47Ala; p.Arg48Ala; p. Ser49Ala). The IL-33-FL constructs and PCR products were digested with the appropriate restriction enzymes (pEGFP-IL-33-FL, HindIII and BbsI; pINDUCER20-IL-33-FL, BsmBI and BstEII). The original restriction fragment excised from each IL-33-FL construct was replaced with the appropriate PCR-generated, digested fragment containing either of the sets of mutations described above. For each construct, Sanger sequencing of the entire replaced restriction fragment was performed to confirm the correct sequence.

**Transduction.** In order to generate TE-7 cells with stable, constitutive overexpression of IL-33, lentivirus was generated by transiently transfecting HEK 293 T cells with either pLVX IL-33 plasmid or empty vector and then transducing TE-7 cells with supernatants in the presence of polybrene (5 μg/mL) with centrifugation at $2000 \times g$ for 1 h at room temperature (RT). The following day, puromycin selection was applied at 1 μg/mL to generate pools of cells with stable overexpression. Following at least 1 week of puromycin treatment, single-cell clones were generated using limiting dilution. In order to generate cells with stable, Dox-inducible overexpression of IL-33, TE-7 cells were transduced with lentivirus pINDUCER20 full-length IL-33–FLAG or truncated (inclusive of a.a. 112–270)

IL-33–FLAG, or the empty vector, as described above except that selection was performed with G418.

**IL-33 induction and RNA sequencing.** To induce IL-33 expression, Dox-inducible TE-7 cells were treated with 100 ng/mL of doxycycline (Clontech) or control media for 48 h. Expression of WT and truncated IL-33 was verified by western blot. RNA was isolated using Tripure reagent and subjected to genome-wide RNA sequencing through the Cincinnati Children's Hospital Medical Center (CCHMC) Gene Expression Core. The RNA-seq results were analyzed using BioWardrobe[53] (http://biowardrobe.cchmc.org). Briefly, the RNA-seq FASTQ files from the Illumina pipeline were aligned by STAR provided with the human RefSeq transcriptome. Differentially expressed genes upon treatment with Dox were identified using differential expression analysis for sequence count data (DESeq)[54]. Venny (http://bioinfogp.cnb.csic.es/tools/venny/) was used to intersect gene lists. For heatmap generation, Cluster 3.0 (http://bonsai.hgc.jp/~mdehoon/software/cluster/software.htm) was used for clustering data using Euclidian distance with average linkage. For visualization, the Java Treeview (http://jtreeview.sourceforge.net/) was used.

**Western blot.** Supernatants were obtained by centrifugation at $3000 \times g$ for 5 min at 4 °C. NuPage LDS loading buffer, beta-mercaptoethanol, and protease inhibitors (Roche) were then added to the supernatants. Cell lysates were extracted from the pellets by lysing cells in RIPA buffer (50 mM Tris-HCl pH 8, 150 mM NaCl, 1% Igepal, 0.5% sodium deoxycholate, 0.1% SDS, and 1 mM EGTA) supplemented with beta-mercaptoethanol and protease inhibitors (Roche) and subsequently sonicating for three rounds of 10 s. Lysates and supernatant samples were boiled for 15 min, loaded onto a 4–12% SDS-PAGE gel (Invitrogen), and subjected to western blot analysis. Membranes were probed with 1:3000 goat anti–IL-33 (AF3625,R&D), 1:10,000 rabbit anti-total p38 MAPK XP (8690, Cell Signaling), 1:2000 rabbit anti-phospho-NFκB (E1Z1T, Cell Signaling), 1:2000 rabbit anti-total NFκB (C22B4, Cell Signaling), 1:3000 goat anti-Lamin B (sc-6217, Santa Cruz), 1:5000 rabbit anti-total histone H3 (ab1791, Abcam), 1:3000 mouse anti-histone H2B (ab52484, Abcam), 1:3000 mouse anti-Hsp90 (TA500494, Origene), or 1:10,000 rabbit anti-GST (Biotin) (ab87834, Abcam) primary antibodies as indicated. Secondary IRDye-conjugated antibodies were from LI-COR Biosciences (Lincoln, Nebraska) and were used at 1:15,000 dilution. Quantification of signal was performed with Image Studio Lite software (http://www.licor.com/bio/products/software/image_studio_lite/). Images have been cropped for presentation purposes. Uncropped, full-size images are present in Supplementary Figures 7, 8, 9.

**Induction of necrosis.** IL-33 expression was induced in pools of Dox-inducible TE-7 cells as above. Cells were treated for 4 h with calcium ionophore A23187 (20 μM) or vehicle. Two sets of supernatants and cell pellets were harvested. Whole-cell lysis was performed on one set of cell pellets as described above; the second set of cell pellets were lysed with 0.5% Triton X-100 in RPMI media. Lactate dehydrogenase activity assay (Promega, catalog # G7890) was performed according to manufacturer's instructions on one set of supernatants and on the 0.5% Triton X-100 cell lysates. western blot was performed on the other set of supernatants and whole-cell lysed cell pellets as described above. In independent experiments, the same pools of TE-7 cells were subjected to two rounds of freeze-thaw (30 min at −80 °C and 1 min at 37 °C per round). Viability was assessed by counts with Trypan blue exclusion, and Western blot was performed on supernatants and cell pellets.

**Biochemical fractionation.** Biochemical fractionation was performed on single-cell clones of TE-7 cells with stable, constitutive overexpression of full-length IL-33 (pLVX IL-33). Briefly, Triton X-100 (0.5%) was used to solubilize both the plasma and nuclear membranes in order to release cytoplasmic, nucleoplasmic, and weakly

**Fig. 8** IL-33 and histones synergistically activate ST2 signaling. **a** Acid-purified histones were run by electrophoresis on an SDS-PAGE gel and subjected to Coommassie blue staining; the right lane depicts the indicated molecular weight markers. **b** Supernatants from HMC-1 mast cells treated overnight with 6.7 μg/mL acid-purified histones or 23.4 nM recombinant wild-type (WT) IL-33 as indicated were subjected to analysis by a 42-cytokine multiplex array. Heat map shows fold change (log10) compared to untreated samples. Bolded cytokines have a ≥2-fold change between samples treated with IL-33 plus histones compared to IL-33 alone. **c** HMC-1 mast cells were treated with anti-ST2 or control IgG for 1 h with subsequent overnight treatment with 6.7 μg/mL of acid-purified histones and 11.7 nM IL-33 as indicated. IL-8 levels in the supernatants were determined by ELISA. **d–f** IL-8 levels in supernatants of HMC-1 mast cells treated overnight with indicated amounts of wild-type IL-33 (**d**), truncated IL-33 (**e**), or IL-1α (**f**) protein in the presence of 6.7 μg/mL of acid-purified histones as indicated. **g–i** Normalization of data from **d–f** depicting fold change as a function of the amount of histones and IL-33 (**g–h**) or IL-1α (**i**) compared to IL-33 or IL-1α treatment alone. Untreated samples and samples receiving only histone are excluded from the graphs. **j** Representative western blot analysis of phospho-NFκB and total NFκB expression in cell lysates of HMC-1 mast cells treated for 20 min with 4.0 μg/mL acid-purified histones or 3.5 nM IL-33 with quantification of proportion of NFκB protein that was phosphorylated in **k**. The data in (**c**) are representative of five independent experiments, **d–i** are representative of 2–3 independent experiments, and (**k**) are cumulative data of three independent experiments. All graphs depict mean ± standard error of the mean. ****, $p < 0.0001$ by Student's $t$-test; *, $p < 0.05$ by one-way ANOVA with Holm–Sidak correction for multiple comparisons. ELISA: enzyme-linked immunosorbent assay, Ig: immunoglobulin, IL interleukin, NFκB: nuclear factor kappa-light-chain-enhancer of activated B cells, SDS-PAGE: sodium dodecyl sulfate polyacrylamide gel electrophoresis, ST2: suppressor of tumorigenicity 2

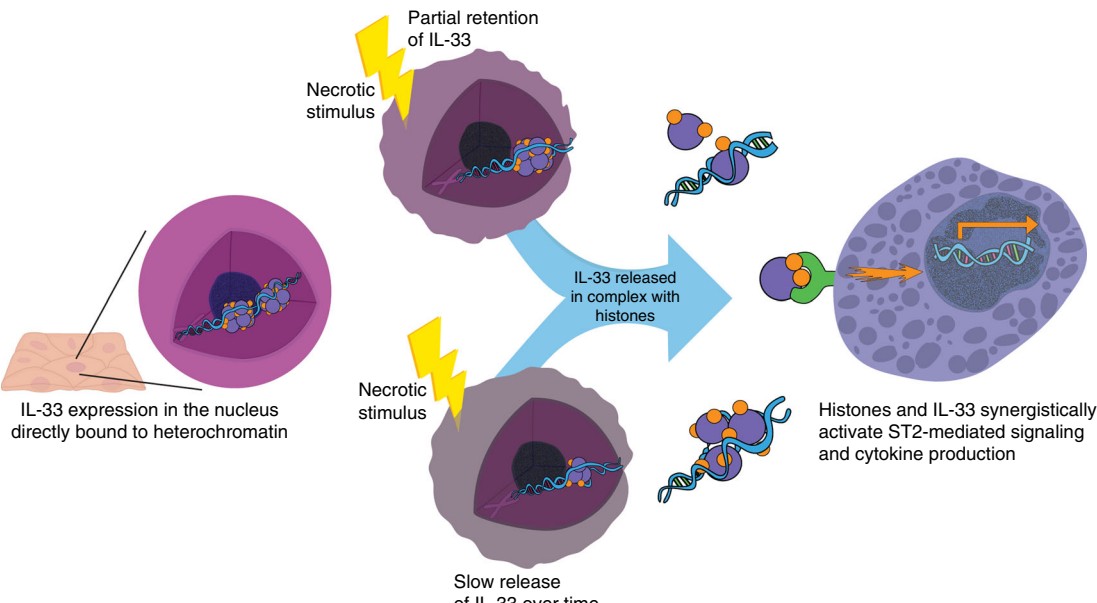

**Fig. 9** Model of regulation of IL-33 cytokine activity by chromatin. In live cells, IL-33 remains within the nuclei of epithelial cells due to IL-33 binding to histones. In response to necrotic stimuli, the integrity of the plasma and nuclear membranes is lost, yielding only partial retention of IL-33. Over time, there is slow release of IL-33 from the necrotic cells, either as a monomer or in complexes with chromatin. Released IL-33 and histones then synergize to induce ST2 signaling and downstream gene expression from ST2-expressing cells. IL, interleukin; ST2, suppressor of tumorigenicity 2

DNA-bound proteins. Proteins with moderate-to-high association with chromatin were then extracted with serial treatment of the residual pellet with micrococcal nuclease (MNase), ammonium sulfate ($(NH_4)_2SO_4$, 0.65 M), and sodium chloride (NaCl, 2 M).

More specifically, cells were washed with 1 mL of cold PBS and aliquoted equally into 1.5-mL Eppendorf tubes. Cells were resuspended in cold Buffer A (10 mM Tris pH 7.5, 100 mM NaCl, 3 mM $MgCl_2$, 1 mM EGTA, and 300 mM sucrose) supplemented with protease inhibitors and 0.5% Triton X-100 and incubated at RT for 5 min in order to permeabilize the cells. The supernatant (S1), representing cytoplasmic and nucleoplasmic proteins, was then collected after centrifugation for 5 min at $350 \times g$. Pellets were then digested with 2000 gel units of micrococcal nuclease (MNase) (NEB) for 20 minu at 37 °C in the presence of 5 mM $CaCl_2$ and 5 mM Tris pH 7.5. Supernatant S2, containing proteins with moderate association with chromatin, was harvested after centrifugation at $2000 \times g$ for 5 min. Residual pellets were then exposed to cold Buffer A supplemented with 0.65 M ammonium sulfate for 15 min at RT to extract remaining chromatin. Supernatant S3, containing proteins with tight association with chromatin, was collected by centrifugation at $2000 \times g$ for 5 min. Pellets were resuspended in 2 M NaCl in $H_20$ and left at RT for 15 min. Supernatant S4 was then harvested following centrifugation at $2000 \times g$ for 5 min. All centrifugation steps were performed at 4 °C. Protein analysis of cell-equivalent volumes of the supernatant and pellet from each fraction was then performed by western blot.

**Live-cell imaging**. TE-7 esophageal epithelial cells were transiently transfected with plasmid encoding GFP-fusion proteins; live-cell imaging experiments were performed 24 h later. For Hoechst co-localization experiments, transfected cells were pre-treated with Hoechst 33342 DNA dye (16.2 mM) for 30 min prior to imaging in PBS supplemented with 10% FBS with a Nikon A1R LUN-V Inverted confocal microscope with a 60×/1.27NA water objective. Pearson correlation coefficients of fluorescence between GFP and Hoechst 33342 within the nucleus of individual cells were determined using Nikon NIS AR elements program.

In separate experiments, GFP fluorescence and differential interference contrast (DIC) were continuously detected. Triton X-100 (final concentration 0.13%) was added to the cells to induce immediate necrosis, and imaging continued for an additional 3 min. The amount of intracellular fluorescence in individual cells was determined over time with normalization to the initial fluorescence in that cell before cell death. Often within the first few seconds of addition of the Triton X-100, the focus of the microscope would be transiently lost due to the physical force of the addition, resulting in a temporary underestimation of the intracellular fluorescence until clear focus resumed a few seconds later.

**Fluorescence recovery after photobleaching**. Briefly, after irreversible laser photobleaching of all GFP-fusion proteins within a nuclear region of interest (ROI), we assessed the recovery of fluorescence within the ROI over time. This recovery occurs because of the influx of other GFP-fusion proteins into that region

that exchange with the photobleached proteins[55]. For chromatin-binding proteins, the amount of time it takes to occur is inversely related to their residence time at binding sites[56]. More specifically, TE-7 esophageal epithelial cells were transiently transfected with plasmid encoding GFP-fusion proteins. After 24 h, FRAP was performed on cells in PBS supplemented with 10% FBS using a Nikon A1R LUN-V Inverted confocal microscope using a 60× /1.27NA water objective and Nyquist zoom. A 15-pixel circular ROI was used to bleach heterochromatic regions defined by intense green staining for three loops of 1 s with 25% percent bleach using the 488-nm laser. Fluorescence within the ROI was recorded at approximately 1-s intervals for the following 60 s. Fluorescence at each timepoint within the ROI was calculated relative to unbleached area with background subtraction and normalized to the pre-bleach signal.

**Procurement and processing of esophageal biopsies**. This study was performed with the approval of the CCHMC Institutional Review Board. Informed consent was obtained from patients or their legal guardians to donate tissue samples for research and to have their clinical information entered into the Cincinnati Center for Eosinophilic Disorders (CCED) database. Patients with active EoE were defined as those having 15 or more eosinophils/HPF at the time of biopsy and not receiving swallowed glucocorticoid or dietary treatment at time of endoscopy. Esophageal biopsies were fixed with formalin and embedded in paraffin (FFPE).

**Immunofluorescence**. Immunofluorescence of esophageal biopsies was performed as previously described[57]. Briefly, slides with 4-µm sections of FFPE esophageal biopsies underwent deparaffinization by serial incubations with xylene, 100% EtOH, 95% EtOH, 70% EtOH, and 50% EtOH. Antigen retrieval was performed using sodium citrate buffer (10 mM sodium citrate, 0.05% Tween 20, pH 6.0). Sections were blocked with blocking buffer (10% donkey serum/PBS) and then incubated with primary antibody directed against IL-33 (mouse clone Nessy-1, ALX-804-840, Enzo, 1:1000 dilution, or goat polyclonal, AF3625, R&D, 1:1000 dilution) or control antibody diluted in blocking buffer overnight at 4 °C in humidified chambers. The next day, slides were washed with PBS, incubated with AlexaFluor 488-conjugate donkey anti-goat IgG or AlexaFluor 647-conjugated donkey anti-mouse IgG secondary antibodies diluted in blocking buffer for 1 h at RT in humidified chambers, and washed in the presence of DAPI (0.5 µg/mL). Finally, a cover slip was added with ProLong Gold mounting reagent (Molecular Probes). Slides were imaged using a Nikon A1R Inverted confocal microscope.

In separate experiments, immunofluorescence was performed on primary esophageal epithelial cells or in single-cell clones of TE-7 cells with stable, constitutive overexpression of full-length IL-33 (pLVX IL-33). Cells were cultured on Ibidi 8-well chambers. Cells were fixed with 4% paraformaldehyde for 10 min, washed with PBS, and permeabilized with 0.1% Triton X-100 in PBS. Cells were blocked with blocking buffer for 30 min and incubated with primary antibodies at 1:500 dilution in blocking buffer for 1–2 h at RT. Cells were washed with PBS, incubated with secondary antibodies diluted in blocking buffer for 1 h at RT, and

washed in the presence of DAPI (0.5 μg/mL). Cells were placed in fresh PBS and imaged using a Nikon A1R Inverted confocal microscope.

**Size-exclusion chromatography**. Necrosis was induced by cryoshock in pools of TE-7 cells with Dox-inducible overexpression of WT or truncated IL-33 as described above. Size-exclusion chromatography was run on supernatants using a GE Healthcare HiPrep 16/60 Sephacryl S-100 HR column. Protein concentration was continuously monitored by measuring UV absorbance at a wavelength of 280 nm. Fractions of 1 mL were collected. IL-33 expression in fractions was determined by IL-33 ELISA (DY3625, R&D). In separate experiments, double-stranded DNA concentration in fractions was quantitated by Qubit (Q32854, Thermo Scientific) following the manufacturer's instructions. Fractions corresponding to elution volumes of 35–39 and 55–59 mL were then pooled, concentrated by acetone precipitation, and subjected to western blot analysis.

**Co-immunoprecipitation**. Necrosis was induced by cryoshock in pools of TE-7 cells with Dox-inducible overexpression of WT IL-33 as described above in PBS. Protein concentration was determined by BCA assay (23227, Thermo Scientific). Equal amounts of proteins were precleared for 2 h at 4 °C with Protein A/G beads (sc-2003, Santa Cruz) in the presence of 250 mM NaCl, 1% NP-40, 1 mM EDTA, and protease inhibitors. Samples were incubated overnight at 4 °C with 2 μg of antibody. Samples were incubated for 1 h at 4 °C with Protein A/G beads before elution with glycine (pH 2.8). Protein expression in eluates was determined by western blot analysis.

In separate experiments, 4 μg each of recombinant GST-tagged full-length IL-33 (H00090865-P01, Abnova) and acid-purified histones were incubated together in 1 mL of PBS, treated with DNase, and incubated for 1 h with mouse anti-histone H2B (ab52484, Abcam) and rabbit anti-histone H2A (07–146, Millipore) or control IgG for 1 h, and then incubated overnight at 4 °C in Protein A/G beads precleared with PBS. Elution was performed with 1 M glycine (pH 2), and protein expression in eluates was determined by western blot analysis.

**Acid extraction of histones**. Histones were isolated from TE-7 cells as described previously[58]. All steps were performed at 4 °C. Briefly, cells were washed with PBS, incubated in hypotonic lysis buffer (10 mM Tris-HCl [pH 8.0], 1 mM KCl, 1.5 mM MgCl$_2$, 0.2% Triton-X 100, 1 mM DTT, protease inhibitors) for 30 min, spun at 10,000 × g for 10 min, and incubated overnight in 0.4 N H$_2$SO$_4$. Supernatants were collected after centrifugation at 16,000 × g for 10 min and incubated for 30 min with trichloroacetic acid (final concentration 33%). Proteins were washed with ice-cold acetone and resuspended in MilliQ water. Protein concentration was determined by BCA assay (23227, Thermo Scientific). Purity of the preparation was assessed by SDS-PAGE and Coomassie blue staining.

**ST2 bioactivity assay**. HMC-1 mast cells were plated in 96-well plates at 50,000 cells per well and stimulated overnight with recombinant full-length human IL-1α (H00003552-PO1, Abnova) recombinant human WT IL-33 (H00090865-P01, Abnova), recombinant human truncated IL-33 (200-33, Peprotech), recombinant histone octamers (31470, Active Motif), or acid-extracted histones prepared as described above. IL-8 levels in supernatants was determined by IL-8 ELISA (431504, BioLegend) following the manufacturer's instructions. In some experiments, mast cells were treated with ST2 blocking antibody (AF523, R&D) or control IgG. In other experiments, supernatants were subjected to 42-cytokine multiplex array (Eve Technologies). Cytokines were considered synergistically regulated when a fold change of combined treatment of IL-33 and histones compared to vehicle-treated cells was higher than the sum of the fold changes of IL-33 alone and histones alone.

In separate experiments, necrosis was induced by cryoshock in pools of TE-7 cells with Dox-inducible overexpression of WT IL-33 cultured at 90% confluency in a 150-mm dish as described above in 1.5 mL of PBS. After collecting 100 uL input, the remaining supernatant was concentrated using a Millipore Amicon Ultra-15 centrifugal filter with a 100 kDa molecular weight cut-off (UFC910024, Millipore), yielding a concentrate of 250 uL. HMC-1 mast cells were then stimulated with dilutions of the input and concentrate as above.

**Statistical analysis**. GraphPad Prism software was used for the indicated statistical analyses. A p value <0.05 was considered to be statistically significant. Student's t-test, one-way analysis of variance (ANOVA), and two-way ANOVA were performed as indicated.

**Data availability statement**. The RNA-seq data have been deposited in NCBI's Gene Expression Omniubus and are accessible through GEO series accession number GSE115097. The datasets generated and/or analyzed during the current study are available from the corresponding author upon reasonable request.

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

## Acknowledgements

The authors wish to thank Shawna Hottinger for editorial assistance and the Cincinnati Children's Confocal Imaging Core. This work was supported by National Institutes of Health R37 AI045898, R01 AI124355, U19 AI070235, T32 GM063483, and F30 DK109573; the Campaign Urging Research for Eosinophilic Disease (CURED); the Buckeye Foundation; and the Sunshine Charitable Foundation and its supporters, Denise A. Bunning and David G. Bunning.

## Author contributions

J.T., M.R., C.E.M., J.E.H., M.B., J.C., and J.K.R. performed experiments and data analysis. J.T., M.R., and C.E.M. wrote the manuscript. M.E.R. supervised the study.

## Additional information

**Competing interests:** M.E.R. is a consultant for Pulm One, Spoon Guru, Celgene, Shire, Astra Zeneca, GlaxoSmithKline, Allakos, Adare, Regeneron and Novartis; has an equity interest in Pulm One, Spoon Guru, and Immune Pharmaceuticals; and has royalties from reslizumab (Teva Pharmaceuticals). M.E.R. is an inventor of several patents owned by Cincinnati Children's. The remaining authors declare no competing interests.

