## [Peer Review File · Nature Communications]

Reviewers' comments:

Reviewer #1 (IL33, ST2)(Remarks to the Author):

In this study, the authors investigated various functional properties of nuclear IL-33. Overexpression of full-length human IL-33, which localized to the cell nucleus, did not affect gene expression in epithelial cells. Interestingly, the authors further report slow intra-nuclear mobility of overexpressed full length IL-33, whereas an N-terminally truncated form of the cytokine, which does not bind to chromatin, was freely mobile. Following necrosis, the chromatin-binding ability of full length IL-33 was associated with decreased release of the cytokine into the extracellular space. Furthermore, when released, full length IL-33 was found together with histone H2B in a high molecular-weight complex also containing DNA, suggesting co-release with chromatin. Notably histones and IL-33 synergistically activated ST2 receptor mediated signaling. Based on their observations, the authors conclude that chromatin binding acts as a posttranslational mechanism to regulate the bioactivity of IL-33.

The manuscript is clear and well-written, previous work is appropriately cited and overall the data are well controlled and solid. The proposed hypothesis is interesting and some, if not all, of the reported findings are novel. The lack of effect of nuclear IL-33 overexpression on gene expression in epithelial cells essentially confirms a previous study performed on a different cell type (reference 31 of submitted manuscript). The N-terminus dependent chromatin-binding ability of full length IL-33 had been already well demonstrated (see for instance references 18 and 21 of submitted manuscript), as well as the fact that a shorter form of IL-33 lacking the N-terminal nuclear localization domain is more easily released (reference 32 of submitted manuscript). In contrast, the various observations pertaining to the dynamics of IL-33 binding to chromatin, the release of the full-length cytokine in a high MW fraction and the synergic activation of ST2-mediated signaling by IL-33 and histones are original and interesting. Nevertheless, several specific issues, as listed below, would deserve to be addressed.

Specific comments:

1. The authors observe a correlation between chromatin binding of full length IL-33 and other functional characteristics, such as enrichment in heterochromatin regions (Figure 3, consistent with previously published data, references 18 and 21 of submitted manuscript) and decreased extracellular release (Figure 5). The authors conclude from these correlations to a causal relationship between chromatin binding and the other observations (p.7, line 152; p.9, line 185; legend Figure 5 p.40, line 754), which in my opinion, is an overstatement. Although this interesting hypothesis may well prove correct, data clearly demonstrating a causal link are in fact not provided here. Indeed the N-terminal truncation used in the present study removes almost half of the protein, including the whole evolutionarily conserved N-terminus, well beyond the CBD. It was shown previously, and confirmed here, that this truncation abolishes not only chromatin binding, but nuclear localization altogether. In order to directly address the role of chromatin binding per se in the studied processes, point mutants specifically abolishing docking to the H2A-H2B dimer (such as the R48A mutant described in reference 21 of the submitted manuscript for instance) should be used to invalidate the CBD. The data obtained using the N-terminal truncation remain of course valid, but the text should be amended to modify the conclusions mentioned above.

2. Methods p.24 and Figure 8: the recombinant human IL-33 used is full length, which is not the most potent bioactive form of IL-33 to induce ST2 dependent responses (see for instance reference 15 of submitted manuscript). In addition, this particular recombinant protein carries an N-terminal GST tag. Its bioactivity is thus clearly expected to be suboptimal, as compared to that of the 'mature' shorter forms likely occurring in vivo. In the context of the experiments shown in Figure 8, panels C to I, it thus seems essential to include recombinant 112-270 human IL-33 as a positive control for maximal bioactivity and to compare its potency to that of full length IL-33 with and without histones. Potential effects of the addition of histones to this short form of IL-33 should also be documented.

3. Figure 8, Panel A: This rather crude histone preparation certainly contains a number of other proteins in addition to the histones, as suggested also by the presence of many minor bands on

the Coomassie stain. The authors may want to comment on this. The authors also need to document whether this preparation contains any remaining DNA. Along the same line, are these additional proteins or DNA required for the effect of the preparation on mast cells? Does the synergy still work using DNase treated samples or using recombinant, rather than purified, histones?

4. Figure 8, Panel C to I: doses are indicated in 'ng', is this meant as ng/ml? Please clarify. 1000 ng(/ml?) of histones seems a high amount. How was this amount chosen? The authors should show a dose response curve.

Also, in comparison, which amount of histones can one expect to find in the supernatant of necrotic cells – after cryoshock where DNA is outside – or in other situations, such as for instance upon loss of membrane integrity - mimicked here by addition of Triton – where (Figure 6) apparently H2B essentially remains intracellular? How 'physiologically' relevant are 1000 ng(/ml?) of histones for 'real life' necrosis situations? The authors may want to comment on these aspects.

5. Methods p.16, line 350 and Figures 3, 4 and 6: why use mouse instead of human IL-1 α when all the rest of the study is performed using human cells and cytokines? This choice should be justified. Can different functional consequences be expected using the mouse instead of the human cytokine? In any case, the fact that mouse IL-1 α was used should be explicitly mentioned again also in the legend of each concerned Figure.

6. Methods p.16-17 and Figures 1 and 5: in different experiments TE-7 cells with stable inducible IL-33 overexpression were used either as single cell clones (Fig 1) or in pools (Fig 5). The authors should shortly explain why they chose to use these alternative approaches for their different types of experiments. Also, are differences expected between the two methods, for instance regarding IL-33 inducibility or expression levels?

7. Methods p.24, line 530 the description/reference for the anti-ST2 antibody is missing.

8. Figure 6 B, C: after an initial decrease, intracellular fluorescence for H2B seems to increase again with time. How can this be explained?

Reviewer #2 (Innate-adaptive crosstalk, ILC)(Remarks to the Author):

Travers et al. examined the significance of nuclear localization and chromosome binding of IL-33. The authors showed that IL-33 binds to chromatin but does not affect overall gene expression in an epithelial cell line. They also showed that IL-33 was released with a slower kinetics than IL-1 α upon cellular necrosis. When the authors used a truncated form (a.a. 112-270) of IL-33 incapable of binding to chromatin, it lost the nuclear localization and was released with much faster kinetics as IL-1 α than a full-length IL-33. The authors further showed that IL-33 was released from necrotic cells as a high molecular form that was not detected when a truncated form was expressed. Lastly, the authors showed a synergistic effect of IL-33 and acid-released histone preparation for the activation of a mast cell line to release more cytokines such as CCL2, CX3CL1 and IL-6 compared to activation by IL-33 alone. From these results, the authors conclude that binding of IL-33 to chromatin does not affect gene expression but controls the release of IL-33 upon necrotic cell death.

This is an interesting topic in the field of allergic inflammation. The authors studied the significance of enigmatic nuclear localization of IL-33 that triggers strong activation of type 2 innate immune responses. Although the topic is of interest and most of the experiments were done in a logical manner, there are several concerns that need to be addressed.

1) The authors observed that IL-33 was released to the supernatant as a high molecular form upon necrotic cells death. It is important to examine if such high molecular complex has an activity to activate cells through ST2. If not, it should be tested if proteases from neutrophils and mast cells change the activity of such high molecular complex.

2) When the authors examined the synergistic effect of IL-33 and histones, the authors used a

truncated version of IL-33 that was incapable of binding to histones. This reviewer does not understand why the authors drew a conclusion described in Fig. 9 that IL-33 and histone form a complex and bind to ST2. The authors should use a full-length IL-33 in this experiment.

3) It is of interest to examine if a full length IL-33 can form a complex with histones in solution. The authors can test this possibility using size exclusion chromatography with a gel filtration column.

4) In Fig. 2B, IL-1 α looks to be localized in the nucleus, which is inconsistent with the data in Fig. 2C and description in the text. The authors should use a better representative picture for this panel.

5) Fig. 7E, histone H2B band in the immunoprecipitates is too faint to see. The authors should use a better picture for this panel.

6) Page 3, 62nd line: "type 2 innate lymphoid cells" should be "group 2 innate lymphoid cells".

Reviewers' comments:

Reviewer #1

1. The authors observe a correlation between chromatin binding of full length IL-33 and other functional characteristics, such as enrichment in heterochromatin regions (Figure 3, consistent with previously published data, references 18 and 21 of submitted manuscript) and decreased extracellular release (Figure 5). The authors conclude from these correlations to a causal relationship between chromatin binding and the other observations (p.7, line 152; p.9, line 185; legend Figure 5 p.40, line 754), which in my opinion, is an overstatement. Although this interesting hypothesis may well prove correct, data clearly demonstrating a causal link are in fact not provided here. Indeed the N-terminal truncation used in the present study removes almost half of the protein, including the whole evolutionarily conserved N-terminus, well beyond the CBD. It was shown previously, and confirmed here, that this truncation abolishes not only chromatin binding, but nuclear localization altogether. In order to directly address the role of chromatin binding per se in the studied processes, point mutants specifically abolishing docking to the H2A-H2B dimer (such as the R48A mutant described in reference 21 of the submitted manuscript for instance) should be used to invalidate the CBD. The data obtained using the N-terminal truncation remain of course valid, but the text should be amended to modify the conclusions mentioned above.

As per the reviewer's request, in order to more definitively demonstrate our claim of a causal relationship between chromatin binding of IL-33 and its properties described in the manuscript, we have generated expression constructs with the suggested mutations within the IL-33 CBD (R48A and 47AAA49). We have found that compared to the full-length protein, these mutant forms of IL-33 have decreased strength of chromatin binding (Supplementary Figure S2), decreased heterochromatic localization (Supplementary Figure S2), increased intranuclear mobility (Supplementary Figure S3), and increased release during necrosis (Supplementary Figures S4 and S5). These data further support our conclusion that chromatin binding regulates IL-33 localization and release.

2. Methods p.24 and Figure 8: the recombinant human IL-33 used is full length, which is not the most potent bioactive form of IL-33 to induce ST2 dependent responses (see for instance reference 15 of submitted manuscript). In addition, this particular recombinant protein carries an N-terminal GST tag. Its bioactivity is thus clearly expected to be suboptimal, as compared to that of the 'mature' shorter forms likely occurring in vivo. In the context of the experiments shown in Figure 8, panels C to I, it thus seems essential to include recombinant 112-270 human IL-33 as a positive control for maximal bioactivity and to compare its potency to that of full length IL-33 with and without histones. Potential effects of the addition of histones to this short form of IL-33 should also be documented.

In response to the reviewer's request, we have performed the suggested experiments with the truncated (a.a. 112-270) IL-33 and have not observed any synergy. These data are now added to Figure 8. We also comment on these new findings in the Discussion.

3. Figure 8, Panel A: This rather crude histone preparation certainly contains a number of other proteins in addition to the histones, as suggested also by the presence of many minor bands on the Coomassie stain. The authors may want to comment on this. The authors also need to document whether this preparation contains any remaining DNA. Along the same line, are these additional proteins or DNA required for the effect of the preparation on mast cells? Does the synergy still work using DNase treated samples or using recombinant, rather than purified, histones?

In response to this request, we have determined that there is a very low concentration (< 0.5 ng/mL) of DNA in the acid-purified histones, demonstrating over 99% removal of DNA from the starting material (Supplementary Figure S6). Additionally, we have reproduced the cooperative activity of histones and full-length IL-33 on ST2 bioactivity using recombinant histone octamers (Supplementary Figure S6).

4. Figure 8, Panel C to I: doses are indicated in 'ng', is this meant as ng/ml? Please clarify. 1000 ng(/ml?) of histones seems a high amount. How was this amount chosen? The authors should show a dose response curve.

We have clarified that we used 1000 ng of histones, which equates to a concentration of $6.7 \mu\text{g/mL}$, for those experiments in Figure 8. Additionally, we have included a dose response involving a broad range of doses of histones in which $6.7 \mu\text{g/mL}$ of histones had the maximal effect (Supplementary Figure S6).

Also, in comparison, which amount of histones can one expect to find in the supernatant of necrotic cells – after cryoshock where DNA is outside – or in other situations, such as for instance upon loss of membrane integrity - mimicked here by addition of Triton – where (Figure 6) apparently H2B essentially remains intracellular? How 'physiologically' relevant are 1000 ng(/ml?) of histones for 'real life' necrosis situations? The authors may want to comment on these aspects.

We have added comments regarding these issues to the Discussion of the manuscript. We cite literature demonstrating the presence of these levels of histones in the sera of human patients and *in vitro* experiments performed by others using similar or even larger concentrations of histones. These data suggest that the amounts of histones used are physiologically relevant.

5. Methods p.16, line 350 and Figures 3, 4 and 6: why use mouse instead of human IL-1 α when all the rest of the study is performed using human cells and cytokines? This choice should be justified. Can different functional consequences be expected using the mouse instead of the human cytokine? In any case, the fact that mouse IL-1 α was used should be explicitly mentioned again also in the legend of each concerned Figure.

We examined the mouse version of IL-1 α because the construct had been used by others to study biochemical properties and nuclear mobility of the protein [Cohen, I. *et al.* Differential release of chromatin-bound IL-1 α discriminates between necrotic and apoptotic cell death by the ability to induce sterile inflammation. *Proc Natl Acad Sci U S A* **107**, 2574-2579, doi:10.1073/pnas.0915018107 (2010)]. We have added this justification and the relevant reference to the Results section when the results from Figure 3 are presented. We do not expect there to be different functional consequences of using the mouse instead of the human cytokine. We have also updated the figure legends to make it clear that a mouse IL-1 α construct was used.

6. Methods p.16-17 and Figures 1 and 5: in different experiments TE-7 cells with stable inducible IL-33 overexpression were used either as single cell clones (Fig 1) or in pools (Fig 5). The authors should shortly explain why they chose to use these alternative approaches for their different types of experiments. Also, are differences expected between the two methods, for instance regarding IL-33 inducibility or expression levels?

Overexpression of IL-33 in the cells presented a substantial challenge in terms of uniformity of protein expression. Therefore, single-cell clones were used in the RNA-sequencing experiment in Figure 1 in order to ensure uniform expression of IL-33 throughout all cells in the population. We utilized pools in order to minimize the potential for artifacts resulting from clonal selection for biochemical experiments. We have added these justifications to the Results section.

7. Methods p.24, line 530 the description/reference for the anti-ST2 antibody is missing.

The reference for this antibody has been added.

8. Figure 6 B, C: after an initial decrease, intracellular fluorescence for H2B seems to increase again with time. How can this be explained?

This initial decrease for H2B occurred because within the first few seconds of addition of Triton X-100, the focus of the microscope is transiently lost due to the physical force of the addition of the Triton. This loss of focus caused a temporary underestimation of the intracellular fluorescence until clear focus resumed a few seconds later. This likely explains why the H2B intracellular fluorescence seems to increase again with time. This explanation has been added to the description of the live-cell imaging experiments in the Methods section.

Reviewer #2

1) The authors observed that IL-33 was released to the supernatant as a high molecular form upon necrotic cells death. It is important to examine if such high molecular complex has an activity to activate cells through ST2. If not, it should be tested if proteases from neutrophils and mast cells change the activity of such high molecular complex.

As per the reviewer's request, we first attempted to measure the ST2 bioactivity of the WT IL-33-containing high molecular weight fractions from the size-exclusion column. However, we could not generate a detectable signal because of the dilution of the sample that occurs during chromatography. Instead, we concentrated necrotic supernatants using a centrifugal filter with a 100 kDa molecular-weight cut-off. Treatment of mast cells with the concentrate obtained after centrifugation induced IL-8 release from mast cells in a dose-dependent fashion and was diminished with an ST2-neutralizing antibody. This data indicates that IL-33 retains its ability to activate the ST2 receptor when it is in complex with chromatin and has been added to the manuscript in Figure 7F.

2) When the authors examined the synergistic effect of IL-33 and histones, the authors used a truncated version of IL-33 that was incapable of binding to histones. This reviewer does not understand why the authors drew a conclusion described in Fig. 9 that IL-33 and histone form a complex and bind to ST2. The authors should use a full-length IL-33 in this experiment.

Binding of IL-33 to histones is required for synergistic activation of mast cells through ST2 binding. However, experiments in Figure 8 were performed with a recombinant full-length form of the protein (containing amino acids 1-270) capable of binding histones. This full-length protein, whose interaction with histones was confirmed by co-immunoprecipitation, induced synergistic activation of mast cells upon co-treatment with acid-purified histones. We now added new data to Figure 8 demonstrating that a truncated form of IL-33 (112-270aa), which is incapable of binding to chromatin, does not synergize with histones in inducing IL-8 release from mast cells.

3) It is of interest to examine if a full length IL-33 can form a complex with histones in solution. The authors can test this possibility using size exclusion chromatography with a gel filtration column.

In response to the reviewer's comment, we performed co-immunoprecipitation of recombinant full-length IL-33 and acid-purified histones. We can detect IL-33 by Western blot after

immunoprecipitation with anti-histone H2A and H2B antibodies but not control antibodies. This data has been added to the manuscript in Supplementary Figure S6.

4) In Fig. 2B, IL-1 α looks to be localized in the nucleus, which is inconsistent with the data in Fig. 2C and description in the text. The authors should use a better representative picture for this panel.

The biochemical fractionation performed in Figure 2A-C separates proteins based on the strength of their interactions with chromatin and/or the nuclear matrix. The protocol does not separate the nucleus from the cytoplasm, so the S1 fraction contains both cytoplasmic and loosely bound nucleoplasmic proteins. Therefore, it is not surprising or contradictory to find some nuclear proteins, such as IL-33 and small amounts of histone H3, in the S1 fraction.

5) Fig. 7E, histone H2B band in the immunoprecipitates is too faint to see. The authors should use a better picture for this panel.

This experiment involved immunoprecipitation with an anti-histone H2B antibody in order to demonstrate a specific interaction between IL-33 and endogenous, instead of overexpressed, H2B. This purpose was achieved as IL-33 was present in the eluates after immunoprecipitation with the anti-H2B antibody as it is visible in the top panel of Figure 7E. There are several reasons for why the H2B band is relatively weak, although still clearly present in the bottom panel. First, necrotic supernatants generated by cryoshock were used as the input, which contain only a small fraction of H2B. Additionally, the immunoprecipitation was performed using cells only expressing endogenous H2B. We have confidence in these results as we have reproduced this finding multiple times, interaction between IL-33 and H2B by co-immunoprecipitation has been reported by others [Roussel, L. et al. Molecular mimicry between IL-33 and KSHV for attachment to chromatin through the H2A-H2B acidic pocket. *EMBO Rep* 9(10):1006-12, doi: 10.1038/embor.2008.145 (2008)], and because we also detected an interaction between recombinant full-length IL-33 and purified histones by co-immunoprecipitation in independent experiments (Supplementary Figure S6B). We have added clarifying text to the Results section of the main text where the results from Figure 7 are presented.

6) Page 3, 62nd line: “type 2 innate lymphoid cells” should be “group 2 innate lymphoid cells”.

The text has been corrected as suggested.

REVIEWERS' COMMENTS:

Reviewer #1 (Remarks to the Author):

The authors have satisfactorily answered all of my questions.

Reviewer #2 (Remarks to the Author):

Travers et al. performed additional experiments and revised the paper. The authors addressed most of the reviewers' concerns and the revisiosn significantly improved the quality of the paper.

REVIEWERS' COMMENTS:

Reviewer #1 (Remarks to the Author):

The authors have satisfactorily answered all of my questions.

Reviewer #2 (Remarks to the Author):

Travers et al. performed additional experiments and revised the paper. The authors addressed most of the reviewers' concerns and the revisiosn significantly improved the quality of the paper.

We thank the reviewers for their efforts and are delighted that they consider our manuscript to have significantly improved.